# On Testing of Samplers *†

**Kuldeep S. Meel**[1]   ⓡ   **Yash Pote** [1]   ⓡ   **Sourav Chakraborty**[2]
[1]School of Computing, National University of Singapore
[2]Indian Statistical Institute, Kolkata

## Abstract

Given a set of items $\mathcal{F}$ and a weight function $\mathtt{wt} : \mathcal{F} \mapsto (0, 1)$, the problem of sampling seeks to sample an item proportional to its weight. Sampling is a fundamental problem in machine learning. The daunting computational complexity of sampling with formal guarantees leads designers to propose heuristics-based techniques for which no rigorous theoretical analysis exists to quantify the quality of generated distributions. This poses a challenge in designing a testing methodology to test whether a sampler under test generates samples according to a given distribution. Only recently, Chakraborty and Meel (2019) designed the first scalable verifier, called Barbarik, for samplers in the special case when the weight function $\mathtt{wt}$ is constant, that is, when the sampler is supposed to sample uniformly from $\mathcal{F}$. The techniques in Barbarik, however, fail to handle general weight functions.

The primary contribution of this paper is an affirmative answer to the above challenge: motivated by Barbarik, but using different techniques and analysis, we design Barbarik2, an algorithm to test whether the distribution generated by a sampler is $\varepsilon$-close or $\eta$-far from any target distribution. In contrast to black-box sampling techniques that require a number of samples proportional to $|\mathcal{F}|$, Barbarik2 requires only $\tilde{O}(tilt(\mathtt{wt}, \varphi)^2/\eta(\eta - 6\varepsilon)^3)$ samples, where the $tilt$ is the maximum ratio of weights of two satisfying assignments. Barbarik2 can handle any arbitrary weight function. We present a prototype implementation of Barbarik2 and use it to test three state-of-the-art samplers.

## 1 Introduction

Motivated by the success of statistical techniques, automated decision-making systems are increasingly employed in critical domains such as medical [19], aeronautics [33], criminal sentencing [20], and military [2]. The potential long-term impact of the ensuing decisions has led to research in the correct-by-construction design of AI-based decision systems. There has been a call for the design of randomized and quantitative formal methods [35] to verify the basic building blocks of the modern AI systems. In this work, we focus on one such core building block: *constrained sampling*.

Given a set of constraints $\varphi$ over a set of variables $X$ and a weight function $\mathtt{wt}$ over assignments to $X$, the problem of constrained sampling is to sample a satisfying assignment $\sigma$ of $\varphi$ with probability proportional to $\mathtt{wt}(\sigma)$. Constrained sampling is a fundamental problem that encapsulates a wide range of sampling formulations [24, 23, 12, 30, 14]. For example, $\mathtt{wt}$ can be used to capture a given

†The authors decided to forgo the old convention of alphabetical ordering of authors in favor of a randomized ordering, denoted by ⓡ. The publicly verifiable record of the randomization is available at https://www.aeaweb.org/journals/policies/random-author-order/search with confirmation code: GH8VZdz4mQIh. For citation of the work, authors request that the citation guidelines by AEA for random author ordering be followed.

prior distribution often represented implicitly through probabilistic models, and $\varphi$ can be used to capture the evidence arising from the observed data, then the problem of constrained sampling models the problem of sampling from the resulting posterior distribution.

The problem of constrained sampling is computationally hard and has witnessed a sustained interest from theoreticians and practitioners, resulting in the proposal of several approximation techniques. Of these, Monte Carlo Markov Chain(MCMC)-based methods form the backbone of modern sampling techniques [3, 7]. The runtime of these techniques depends on the length of the random walk, and the Markov chains that require polynomial walks are called rapidly mixing Markov chains. Unfortunately, for most distributions of practical interest, it is infeasible to design rapidly mixing Markov chains [26], and the practical implementations of such techniques have to resort to the usage of heuristics that violate theoretical guarantees. The developers of such techniques, often and rightly so, strive to demonstrate their effectiveness via empirical behavior in practice [6].

The need for the usage of heuristics to achieve scalability is not restricted to just MCMC methods but is widely observed for other methods such as simulated annealing [29], variational methods [18], and hashing-based techniques [12, 23, 13, 32]. Consequently, a fundamental problem for the designers of sampling techniques is: *how can one efficiently test whether a given technique samples from the desired distribution?* Most of the existing approaches rely on the computations of statistical metrics such as variation distance and KL-divergence by drawing samples and perform hypothesis testing with a preset p-value. Sound computations of statistical metrics require a large number of samples that is proportional to the support of the posterior distribution [4, 36], which is prohibitively large; it is not uncommon for the distribution support to be significantly larger than $2^{70}$. Consequently, the existing approaches tend to estimate the desired quantities using a fraction of the required samples, and such estimates are often without the required confidence. The usage of unsound metrics may lead to unsound conclusions, as demonstrated by a recent study where the usage of unsound metric would lead one to conclude that two samplers were indistinguishable (it is worth mentioning that the authors of the study clearly warn the reader about the unsoundness of the underlying metrics) [21].

The researchers in the sub-field of property testing within theoretical computer science have analyzed the sample complexity of testing under different models of samplers and computation. The resulting frameworks have not witnessed widespread adoption to practice due to a lack of samplers that can precisely fit the models under which results are obtained. In recent work, Chakraborty and Meel [10], building on the concepts developed in the condition sampling model (rf. [1]), designed the first practical algorithmic procedure, called Barbarik, that can rigorously test whether a given sampler samples from the uniform distribution using a constant number of samples, assuming that the given sampler is *subquery-consistent* (see Definition 9). Empirically, Barbarik was shown to be able to distinguish samplers that were indistinguishable in prior studies based on unsound metrics. While Barbarik made significant progress, it is marred by its ability to handle only the uniform distribution. Therefore, one wonders: *Can we design an algorithmic framework to test whether the distribution generated by a given sampler is close to a desired (but arbitrary) posterior distribution of interest?*

This paper's primary contribution is the first efficient algorithmic framework, Barbarik2, to test whether the distribution generated by a sampler is $\varepsilon$-close or $\eta$-far from the desired distribution specified by the set of constraints $\varphi$ and a weight function wt. In contrast to the statistical techniques that require an exponential or sub-exponential number of samples for samplers whose support can be represented by $n$ bits, the number of samples required by Barbarik2 depends on the *tilt* of the distribution, where *tilt* is defined as the maximum ratio of non-zero weights of two solutions of $\varphi$. Like Barbarik, the key technical idea of Barbarik2 sits at the intersection of *property testing* and *formal methods* and uses ideas from conditional sampling and employs chain formulas. However, the key algorithmic framework of Barbarik2 differs significantly from Barbarik, and, as demonstrated, the proof of its correctness and sample complexity requires an entirely new set of technical arguments.

Given access to an ideal sampler $\mathcal{A}$, Barbarik2 accepts every sampler that is $\varepsilon$-close to $\mathcal{A}$ while its ability to reject a sampler that is $\eta$-far from $\mathcal{A}$ assumes that the sampler under test is *subquery consistent*. Since Barbarik2 assumes access to an ideal sampler, one might wonder if a tester such as Barbarik2 is needed when we already have access to an ideal sampler. Since sampling is computationally intractable, it is almost always the case that an ideal sampler $\mathcal{A}$ is quite slow and one would prefer to use some other efficient sampler $\mathcal{G}$ instead of $\mathcal{A}$, if $\mathcal{G}$ can be certified to be close to $\mathcal{A}$.

To demonstrate the practical efficiency of Barbarik2, we developed a prototype implementation in Python and performed an experimental evaluation with several samplers. While our framework

does not put a restriction on the representation of wt, we perform empirical validation with weight distributions corresponding to log-linear models, a widely used class of distributions. Our empirical evaluation shows that Barbarik2 returns ACCEPT for the samplers with formal guarantees but returns REJECT for other samplers that are without formal guarantees. Our ability to reject samplers provides evidence in support of our assumption of subquery consistency of samplers. We believe our formalization of testing of samplers and the design of the algorithmic procedure, Barbarik2, contributes to the design of *randomized formal methods* for verified AI, a principle argued by Seshia et al [35].

## 2 Notations and Preliminaries

A Boolean variable is denoted by a lowercase letter. For a Boolean formula $\varphi$, the set of variables appearing in $\varphi$, called the *support* of $\varphi$, is denoted by $Supp(\varphi)$. An assignment $\sigma \in \{0,1\}^{|Supp(\varphi)|}$ to the variables of $\varphi$ is a *satisfying assignment* or *witness* if it makes $\varphi$ evaluate to 1. We denote the set of all satisfying assignments of $\varphi$ as $R_\varphi$. For $S \subseteq Supp(\varphi)$, we use $\sigma_{\downarrow S}$ to indicate the projection of $\sigma$ over the set of variables in $S$. And we denote by $R_{\varphi \downarrow S}$ the set $\{\sigma_{\downarrow S} \mid \sigma \in R_\varphi\}$.

**Definition 1** (Weight Function). *For a set $S$ of Boolean variables, a weight function* wt $: \{0,1\}^{|S|} \to (0,1)$ *maps each assignment to some weight.*

**Definition 2** (Sampler). *A sampler $\mathcal{G}(\varphi, S, \text{wt}, \tau)$ is a randomized algorithm that takes in a Boolean formula $\varphi$, a weight function* wt, *a set $S \subseteq Supp(\varphi)$ and a positive integer $\tau$ and outputs $\tau$ independent samples from $R_{\varphi \downarrow S}$. For brevity of notation we will omit arguments $\varphi, S, \text{wt}, \tau$, whenever may sometimes refer to a sampler as $\mathcal{G}(\varphi)$ or simply, $\mathcal{G}$.*

*For any $\sigma \in \{0,1\}^{|S|}$ the probability of the sampler $\mathcal{G}$ outputting $\sigma$ is denoted by $p_\mathcal{G}(\varphi, S, \sigma)$ (or $p_\mathcal{G}(\varphi, \sigma)$ when the set $S$ in question is clear from the context).*

We use $D_{\mathcal{G}(\varphi,S)}$ to represent the distribution induced by $\mathcal{G}(\varphi, S)$ on $R_{\varphi \downarrow S}$. When the set $S$ is understood from the context we will denote $D_{\mathcal{G}(\varphi,S)}$ by $D_{\mathcal{G}(\varphi)}$.

**Definition 3** (Ideal Sampler). *For a weight function* wt, *a sampler $\mathcal{A}(\varphi, S, \tau)$ is called an ideal sampler w.r.t. weight function* wt *if for all $\sigma \in R_{\varphi \downarrow S}$: $p_\mathcal{A}(\varphi, S, \text{wt}, \sigma) = \frac{\text{wt}(\sigma)}{\sum_{\sigma' \in R_{\varphi \downarrow S}} \text{wt}(\sigma')}$. In the rest of the paper, $\mathcal{A}(\cdot, \cdot, \cdot, \cdot)$ denotes the ideal sampler. When* wt$(\sigma) = \frac{1}{|R_\varphi|}$ *then the ideal sampler is called a uniform sampler.*

**Definition 4** (Tilt). *For a Boolean formula $\varphi$ and weight function* wt, *we define $tilt(\text{wt}, \varphi) = \max\limits_{\sigma_1, \sigma_2 \in R_\varphi} \frac{\text{wt}(\sigma_1)}{\text{wt}(\sigma_2)}$.*

Our goal is to design a program that can test the quality of a sampler with respect to an ideal sampler. We use two different notions of distance of the sampler from the ideal sampler.

**Definition 5** ($\varepsilon$-closeness and $\eta$-farness). *A sampler $\mathcal{G}$ is $\varepsilon$-multiplicative-close (or simply $\varepsilon$-close) to an ideal sampler $\mathcal{A}$, if for all $\varphi$ and all $\sigma \in R_\varphi$, we have $(1-\varepsilon)p_\mathcal{A}(\varphi, \sigma) \leq p_\mathcal{G}(\varphi, \sigma) \leq (1+\varepsilon)p_\mathcal{A}(\varphi, \sigma)$. For a formula $\varphi$, a sampler $\mathcal{G}(\varphi)$ is $\eta$-$\ell_1$-far (or simply $\eta$-far) from the ideal sampler $\mathcal{A}(\varphi)$, if $\sum_{\sigma \in R_\varphi} |p_\mathcal{A}(\varphi, \sigma) - p_\mathcal{G}(\varphi, \sigma)| \geq \eta$*

It is worth emphasising that the asymmetry in the notions of $\varepsilon$-close and $\eta$-far stems from the availability of practical samplers. Since the available off-the-shelf solvers with theoretical guarantees provide the guarantee of $\varepsilon$-closeness, we are interested in accepting a sampler that is $\varepsilon$-close [24, 23, 12, 14]. On the other hand, we would like to be more forgiving to the samplers without guarantees and would like to reject only if they are $\eta$-far in $\ell_1$ distance, a notion more relaxed than multiplicative closeness.

**Definition 6** (($\varepsilon, \eta, \delta$)-tester for samplers). *A $(\varepsilon, \eta, \delta)$-tester for samplers is a randomized algorithm that takes a sampler $\mathcal{G}$, an ideal sampler $\mathcal{A}$, a tolerance parameter $\varepsilon$, an intolerance parameter $\eta$, a guarantee parameter $\delta$ and a CNF formula $\varphi$ such that (1) If $\mathcal{G}(\varphi)$ is $\varepsilon$-close to $\mathcal{A}(\varphi)$, then the tester returns ACCEPT with probability at least $(1-\delta)$, and (2) If $\mathcal{G}(\varphi)$ is $\eta$-far from $\mathcal{A}(\varphi)$ then the tester returns REJECT with probability at least $(1-\delta)$.*

## 2.1 Chain Formula

A crucial component in our algorithm is the chain formula. Chain formulas, introduced in [15], are a special class of Boolean formulas. Given a positive integer $k$ and $m$, chain formulas provide an efficient construction of a Boolean formula $\psi_{k,m}$ with exactly $k$ satisfying assignments with $\lceil log(k) \rceil \leq m$ variables. We employ chain formulas for inverse transform sampling and in the subroutine Barbarik2Kernel.

**Definition 7.** *[15] Let $c_1 c_2 \cdots c_m$ be the $m$-bit binary representation of $k$, where $c_m$ is the least significant bit. We then construct a chain formula $\varphi_{k,m}(\cdot)$ on $m$ variables $a_1, \ldots a_m$ as follows. For every $j$ in $\{1, \ldots m-1\}$, let $C_j$ be the connector "$\vee$" if $c_j = 1$, and the connector "$\wedge$" if $c_j = 0$. Define*

$$\varphi_{k,m}(a_1, \cdots a_m) = a_1\, C_1\, (a_2\, C_2 (\cdots (a_{m-1}\, C_{m-1}\, a_m) \cdots))$$

For example, consider $k = 11$ and $m = 4$. The binary representation of 11 using 4 bits is 1011. Therefore, $\varphi_{5,4}(a_1, a_2, a_3, a_4) = a_1 \vee (a_2 \wedge (a_3 \vee a_4))$.

**Lemma 1.** *[15] Let $m > 0$ be a natural number, $k < 2^m$, and $\varphi_{k,m}$ as defined above. Then $|\varphi_{k,m}|$ is linear in $m$ and $\varphi_{k,m}$ has exactly $k$ satisfying assignments. Every chain formula $\psi$ on $n$ variables is equivalent to a CNF formula $\psi^{CNF}$ having at most $n$ clauses. In addition, $|\psi^{CNF}|$ is in $O(n^2)$.*

## 2.2 Barbarik2Kernel and the Subquery Consistency Assumption

Barbarik2Kernel is a crucial subroutine that we use in our algorithm to help us draw *conditional samples* from $R_{\varphi \downarrow S}$. This is similar to the subroutine Kernel used by the Barbarik in [10]. We will now define a collection of functions KernelFamily.

**Definition 8.** KernelFamily *is family of functions that take a Boolean formula $\varphi$, a set of variables $S \subseteq Supp(\varphi)$, and two assignments $\sigma_1, \sigma_2 \in R_{\varphi \downarrow S}$, and return $\hat{\varphi}$ such that $R_{\hat{\varphi} \downarrow S} = \{\sigma_1, \sigma_2\}$.*

[10] introduced the notion of *non-adversarial assumption*, which was crucial in their analysis. We rename the notion of *subquery consistency* to better capture its intended properties, defined below.

**Definition 9.** *Let* Barbarik2Kernel $\in$ KernelFamily. *A sampler $\mathcal{G}$ is subquery consistent w.r.t. a particular* Barbarik2Kernel *for $\varphi$ if for all $S \subseteq Supp(\varphi)$, $\sigma_1, \sigma_2 \in R_{\varphi \downarrow S}$, let $\hat{\varphi} \leftarrow$* Barbarik2Kernel$(\varphi, S, \sigma_1, \sigma_2)$ *then the output of $\mathcal{G}(\hat{\varphi}, \mathtt{wt}, S, \tau)$ is $\tau$ independent samples from the conditional distribution $\mathcal{D}_{\mathcal{G}(\varphi)|T}$, where $T = \{\sigma_1, \sigma_2\}$.*

Similar to the usage of *non-adversarial assumption* in the correctness analysis of Barbarik [10], the notion of *subquery consistency* would play a crucial role in our analysis. Since each subquery can be viewed as conditioning and given that conditioning is a fundamental operation, one would expect that off the shelf samplers would be subquery consistent. At the same time, in contrast to practical applications, the set $T$ is arbitrarily chosen, and therefore, it is possible that certain samplers do not satisfy the property of subquery consistency. It is, however, not known how to test whether a sampler is subquery consistent w.r.t a particular Barbarik2Kernel. While our empirical evaluation provides weak evidence to our claim that off the shelf samplers are subquery consistent, we believe checking whether a sampler is subquery consistent is an interesting and important problem for future work.

## 3 Related Work

Distribution testing involves testing whether an unknown probability distribution is identical or close to a given distribution. This problem has been studied extensively in the property testing literature [11, 8, 36, 37]. The sample space is exponential, and for many fundamental distributions, including uniform, it is prohibitively expensive in terms of samples to verify closeness. This led to the development of the conditional sampling model [11, 8], which can provide sub-linear or even *constant* sample complexities for the testing of the above-given properties[1, 28, 5, 9, 17]. A detailed discussion on prior work in property testing and their relationship to Barbarik2 is given in Appendix A.

The first practically efficient algorithm for verification of samplers with a formal proof of correctness was presented by Chakraborty and Meel in form of Barbarik [10]. The central idea of Barbarik, building on the work of Chakraborty et al. [11] and Canonne et al. [8], was that if one can have

conditional samples from the distribution, then one can test properties of the distribution using only a constant number of conditional samples.

Barbarik constructs a two-element set $T \subset R_\varphi$, with one element drawn according to the distribution $D_{\mathcal{G}(\varphi)}$ and one element drawn uniformly at random from the set $R_\varphi$. Using a subroutine Kernel Chakraborty et al. argued that one can draw samples from the conditional distribution $D_{\mathcal{G}(\varphi)|T}$. Their sample complexity was $\tilde{O}(1/(\eta - 2\varepsilon)^4)$. They proved that if a sampler $\mathcal{G}$ is $\varepsilon$-close to a uniform sampler then Barbarik will accept with probability at least $(1 - \delta)$, while if $\mathcal{G}(\varphi)$ is $\eta$-far from the uniform sampler and if $\mathcal{G}$ is subquery consistent w.r.t Kernel for $\varphi$, then Barbarik rejects with probability at least $(1 - \delta)$. Their underlying assumption was that many samplers that are in use would in fact be *subquery consistent* and the success of Barbarik in rejecting several samplers provides evidence in support of the aforementioned assumption. They used Barbarik to test the correctness of samplers like STS, Quicksampler, and UniGen.

Note that Barbarik can only distinguish a uniform sampler from a far-from uniform sampler, and the techniques used cannot be generalized easily to the case where the ideal sampler is not necessarily uniform. While Barbarik2, that we present in this paper, does borrow several techniques from Barbarik, including drawing inspiration from the concept of conditional sampling for their design; Barbarik2 is very different from Barbarik both in terms of the algorithmic design and its implementation.

# 4  An overview of the Barbarik2 Algorithm

In this section, we present the algorithmic framework of Barbarik2, the pseudocode, presented as Algorithm 1, and then the theoretical justification for the algorithm. Barbarik2 takes as input a blackbox sampler $\mathcal{G}$, a Boolean formula $\varphi$ with the associated weight function wt and three parameters $(\varepsilon, \eta, \delta)$. It also has access to an ideal sampler $\mathcal{A}$. Barbarik2 is an $(\varepsilon, \eta, \delta)$-tester for samplers. Also if Barbarik2 returns REJECT (that is, when $\mathcal{G}$ is $\eta$-far from $\mathcal{A}$), it provides as witness a new formula $\hat{\varphi}$ which is similar to $\varphi$, except that $\hat{\varphi}$ has only two assignments to the variables in $S$ (namely $\sigma_1$ and $\sigma_2$) that can be extended to satisfying assignments of $\hat{\varphi}$ and the relative probability masses of $\sigma_1$ and $\sigma_2$ in $\mathcal{D}_\mathcal{G}$ are significantly different from that in $\mathcal{D}_\mathcal{A}$.

The core idea of Barbarik2 is that for verifying the quality of the sampler $\mathcal{G}(\varphi)$, we can proceed in two stages. In the first stage, if the sampler is far from the ideal sampler $\mathcal{A}$, we hope to find a witness (in the form of two satisfying assignments) for farness with good probability. This can be guaranteed by drawing one sample each from $\mathcal{D}_{\mathcal{G}(\varphi)}$ and $\mathcal{D}_{\mathcal{A}(\varphi)}$. In the second stage, we confirm whether the witness is indeed far. That is, if the witness is the $(\sigma_1, \sigma_2)$ pair, we check that the probability of $\sigma_1$ and $\sigma_2$ in $\mathcal{D}_{\mathcal{G}(\varphi)}$ and $\mathcal{D}_{\mathcal{A}(\varphi)}$ are similar or not.

Here Barbarik2 differs from Barbarik in a significant way. Barbarik employs a bucketing strategy. But, Barbarik2 chooses a simpler yet equally effective method to check the similarity between $\sigma_1$ and $\sigma_2$. This is also the most difficult stage of the tester as one may have to draw a exponential number of samples to confirm the similarity. We manage this by drawing samples from the conditional distribution $\mathcal{D}_{\mathcal{G}(\varphi)|\{\sigma_1,\sigma_2\}}$ instead of $\mathcal{D}_{\mathcal{G}(\varphi)}$. Since $\mathcal{D}_{\mathcal{G}(\varphi)|\{\sigma_1,\sigma_2\}}$ is supported on a set of size only two estimating the distance of $\mathcal{D}_{\mathcal{G}(\varphi)|\{\sigma_1,\sigma_2\}}$ from $\mathcal{D}_{\mathcal{A}(\varphi)|\{\sigma_1,\sigma_2\}}$ can be done with constant number of samples.

Now since we do not have direct access to the distribution $\mathcal{D}_{\mathcal{G}(\varphi)|\{\sigma_1,\sigma_2\}}$ we circumvent the problem by drawing samples from a new distribution $\mathcal{D}_{\mathcal{G}(\hat{\varphi})}$ where $\hat{\varphi}$ is obtained from $\varphi$ and has similar structure as $\varphi$ (with $Supp(\varphi) \subseteq Supp(\hat{\varphi})$) and there are only two assignments (namely $\sigma_1$ and $\sigma_2$) to the variables in $Supp(\varphi)$ that can be extended to the satisfying assignments of $\hat{\varphi}$. The subroutine Barbarik2Kernel helps us simulate the drawing of samples from $\mathcal{D}_{\mathcal{G}(\varphi)|\{\sigma_1,\sigma_2\}}$ by drawing samples from $\mathcal{D}_{\mathcal{G}(\hat{\varphi})}$. The subroutine Bias helps to estimate the distance of $\mathcal{D}_{\mathcal{G}(\hat{\varphi})}$ from $\mathcal{D}_{\mathcal{A}(\hat{\varphi})}$.

Finally, we repeat the whole process for a certain number of rounds, and we argue that if the sampler is indeed far then, with high probability, in at least one round, we will find a witness of farness and confirm that the witness is indeed far. On the other hand, if the sampler is close to ideal, then there does not exist any such witness of farness.

Barbarik2 accesses two subroutines, Bias and Barbarik2Kernel: $\text{Bias}(\hat{\sigma}, \Gamma, S)$ takes as input an assignment $\hat{\sigma}$, a list $\Gamma$ of assignments and a sampling set $S$. It returns the fraction of assignments of $\Gamma$ whose projections on $S$ is equal to $\hat{\sigma}$. $\text{Barbarik2Kernel}(\varphi, \sigma_1, \sigma_2)$ is a Barbarik2Kernel subroutine

**Algorithm 1** Barbarik2($\mathcal{G}, \mathcal{A}, \varepsilon, \eta, \delta, \varphi, S, \mathtt{wt}$)

1: $t \leftarrow ln(1/\delta)ln\left(\frac{10}{10-\eta(\eta-6\varepsilon)}\right)^{-1}$
2: $n \leftarrow 8ln\,(t/\delta)$
3: $lo = (1+\varepsilon)/(1-\varepsilon)$
4: $hi = 1 + (\eta + 6\varepsilon)/4$
5: $\Gamma_1 \leftarrow \mathcal{G}(\varphi, S, t)$;
6: $\Gamma_2 \leftarrow \mathcal{A}(\varphi, S, t)$;
7: **for** $i = 1$ to $t$ **do**
8: $\quad \sigma_1 \leftarrow \Gamma_1[i]$; $\sigma_2 \leftarrow \Gamma_2[i]$;
9: $\quad$ **if** $\sigma_1 = \sigma_2$ **then**
10: $\quad\quad$ **continue**
11: $\quad \alpha \leftarrow \mathtt{wt}(\sigma_1)/\mathtt{wt}(\sigma_2)$
12: $\quad L \leftarrow (\alpha \cdot lo)\,/\,(1 + \alpha \cdot lo)$
13: $\quad H \leftarrow (\alpha \cdot hi)\,/\,(1 + \alpha \cdot hi)$
14: $\quad T = (H + L)/2$
15: $\quad N \leftarrow n \cdot H/(H - L)^2$
16: $\quad \hat{\varphi} \leftarrow$ Barbarik2Kernel$(\varphi, \sigma_1, \sigma_2)$
17: $\quad \Gamma_3 \leftarrow \mathcal{G}(\hat{\varphi}, S, N)$
18: $\quad Bias \leftarrow$ Bias$(\sigma_1, \Gamma_3, S)$
19: $\quad$ **if** $Bias > T$ **then**
20: $\quad\quad$ **return** REJECT
21: **return** ACCEPT

**Algorithm 2** Barbarik2Kernel($\varphi, \sigma_1, \sigma_2$)

1: $m \leftarrow 12, k \leftarrow 2^m - 1$
2: $\mathbf{Lits_1} \leftarrow (\sigma_1 \setminus \sigma_2)$
3: $\mathbf{Lits_2} \leftarrow (\sigma_2 \setminus \sigma_1)$
4: $\mathbf{V} \leftarrow NewVars(\varphi, m)$;
5: $\hat{\varphi} \leftarrow \varphi \wedge (\sigma_1 \vee \sigma_2)$
6: $l \sim \mathbf{Lits_1} \cup \mathbf{Lits_2}$
7: $\hat{\varphi} \leftarrow \hat{\varphi} \wedge (\neg l \to \psi_{k,m}(\mathbf{V}))$
8: $\hat{\varphi} \leftarrow \hat{\varphi} \wedge (l \to \psi_{k,m}(\mathbf{V}))$
9: **return** $\hat{\varphi}$

**Algorithm 3** Bias($\hat{\sigma}, \Gamma, S$)

1: $count = 0$
2: **for** $\sigma \in \Gamma$ **do**
3: $\quad$ **if** $\sigma_{\downarrow S} = \hat{\sigma}$ **then**
4: $\quad\quad$ count $\leftarrow$ count $+1$
5: **return** $\frac{count}{|\Gamma|}$

(Definition 8). Its aim is to create a $\hat{\varphi}$ such the behaviour of the sampler on $\hat{\varphi}$ is similar to it's behaviour on $\varphi$, i.e. $\mathcal{D}_{\mathcal{G}(\varphi)|\{\sigma_1, \sigma_2\}} \approx \mathcal{D}_{\mathcal{G}(\hat{\varphi})}$.

In Barbarik2, in the for loop (in lines $7-20$), in each round, the algorithm draws one sample $\sigma_1$ according to the distribution $\mathcal{D}_{\mathcal{G}(\varphi)}$ and one sample $\sigma_2$ according to the ideal distribution on $R_\varphi$ (line 8). In the case that $\sigma_1 = \sigma_2$ it moves the to next iteration (in line 9-10). In line 16, the subroutine Barbarik2Kernel uses $\varphi$, the two samples $\sigma_1$ and $\sigma_2$, to output a new formula $\hat{\varphi}$ such that $Supp(\varphi) \subseteq Supp(\hat{\varphi})$. On line 17, Barbarik2 draws a list, $\Gamma_3$, of $N$ samples according to the distribution $\mathcal{D}_{\mathcal{G}(\hat{\varphi})}$. Barbarik2Kernel ensures that for all $\sigma \in \Gamma_3$, $\sigma_{\downarrow S}$ is either $\sigma_1$ or $\sigma_2$. In line 18 Barbarik2 uses Bias to compute the fraction of samples that are equal to $\sigma_1$ (on the variable set $S$), and if the fraction is greater than the threshold then Barbarik2 returns REJECT (in line 20).

Algorithm 2 presents the pseudocode of subroutine Barbarik2Kernel. As stated above, Barbarik2Kernel takes in a Boolean formula $\varphi$, a set $S \subseteq Supp(\varphi)$ and two partial assignments $\sigma_1, \sigma_2 \in R_{\varphi \downarrow S}$ . Since the set $S$ is implicit from $\sigma_1$ and $\sigma_2$ it may not be explicitly given as an input. Barbarik2Kernel assumes access to subroutine $NewVars$ which takes in two parameters, a formula $\varphi$ and a number $m$, and returns a set of $m$ fresh variables that do not appear in $\varphi$. Barbarik2Kernel first constructs two sets of literals, denoted by $\mathbf{Lits_1}$ (resp. $\mathbf{Lits_2}$), which appear in $\sigma_1$ (resp. $\sigma_2$) but not $\sigma_2$ (resp. $\sigma_1$). The algorithm then constructs the formula $\hat{\varphi}$. First it generates $\varphi \wedge (\sigma_1 \vee \sigma_2)$ on Line 5, a formula with exactly two solutions. Next, it randomly chooses a literal $l$ from $\mathbf{Lits_1} \cup \mathbf{Lits_2}$ and constructs a chain formula $(l \to \psi_{k,m})$ over the fresh Boolean variables $\mathbf{V}[1], \mathbf{V}[2] \cdots, \mathbf{V}[m]$ where $k$ is the number of satisfying assignments the formula has. Conjuncting the two generated formulas, we get $\hat{\varphi} \equiv \varphi \wedge (\sigma_1 \vee \sigma_2)$. Therefore, at the end of Barbarik2Kernel, i.e. line 8, $\hat{\varphi}$ has $2k$ solutions. We choose the value of $k$ such that it is odd (see [15]). The chain formula is linked to a random Boolean literal from the given set of literals for two reasons,

1. An ideal or $\varepsilon$-close to ideal sampler would not be affected by the randomization and would generate the same distribution over $\hat{\varphi}$ as it does over $\varphi \wedge (\sigma_1 \vee \sigma_2)$.

2. If the sampler under test $\mathcal{G}$ is $\eta$-far from ideal, then we want to construct a formula which *cannot* be easily guessed by $\mathcal{A}$. We wish to avoid the scenario where $\mathcal{A}$, an $\eta$-far sampler on $\varphi$, somehow behaves as an almost-ideal sampler over $\hat{\varphi}$ and hence manages to fool Barbarik2.

## 4.1 Theoretical Analysis

The following theorem gives the mathematical guarantee about the correctness of Barbarik2.

**Theorem 1.** *Given sampler $\mathcal{G}$, ideal sampler $\mathcal{A}$, $\varepsilon < \frac{1}{3}$, $\eta > 6\varepsilon$, $\delta$, $\varphi$ and weight function* `wt`*, Barbarik2 needs at most* $\widetilde{O}\left(\frac{tilt(\mathtt{wt},\varphi)^2}{\eta(\eta - 6\varepsilon)^3}\right)$ *samples, where $\widetilde{O}$ hides a poly logarithmic factor of $1/\delta$.*

- *If $\mathcal{G}$ is an $\varepsilon$-close to $\mathcal{A}$ then* Barbarik2 *returns* `ACCEPT` *with probability at least $(1 - \delta)$.*
- *If $\mathcal{G}$ is subquery consistent w.r.t* Barbarik2Kernel *and if the distribution $\mathcal{D}_{\mathcal{G}(\varphi)}$ is $\eta$-far from the ideal sampler then* Barbarik2 *returns* `REJECT` *with probability at least $(1 - \delta)$.*

Note that if $\mathcal{G}$ is $\varepsilon$-close to $\mathcal{A}$ then Barbarik2 accepts (with high probability) even if the sampler $\mathcal{G}$ isn't subquery consistent w.r.t Barbarik2Kernel. It is also worth noting that Barbarik2 terminates with `REJECT` as soon as the check in line 19 succeeds. Therefore, we expect Barbarik2 to require significantly less number of samples when it returns `REJECT`. Furthermore, in the case of `ACCEPT`, the bound on $N$, as calculated on line 15 in terms of $tilt$, is pessimistic as the probability of observing $\sigma_1$ and $\sigma_2$ such that $\alpha \approx tilt$ for a sampler close to ideal is very small when the tilt is large. The proof of Theorem 1 is presented in Appendix B.

## 5 Evaluation

The objective of our evaluation was to answer the following questions:

**RQ1.** Is Barbarik2 able to distinguish between off-the-shelf samplers by returning `ACCEPT` for samplers $\varepsilon$-close to the ideal distribution and `REJECT` for the $\eta$-far samplers?

**RQ2.** What improvements do we observe over the baseline?

**RQ3.** How does the required number of samples scale with the $tilt(\mathtt{wt}, \varphi)$ of the distribution?

To evaluate the runtime performance of Barbarik2 and test the quality of some state of the art samplers, we implemented a prototype of Barbarik2 in Python. Our algorithm utilizes an ideal sampler, for which we use the state of the art sampler WAPS [25]. All experiments were conducted on a high performance computing cluster with 600 E5-2690 v3 @2.60GHz CPU cores. For each benchmark, we use a single core with a timeout of 24 hours. The detailed logs along with list of benchmarks and the runtime code employed to run the experiments are available at `http://doi.org/10.5281/zenodo.4107136`.

We focus on the log-linear distributions given their ubiquity of usage in machine learning; a formal description is provided in Appendix C for completeness. Observe that Barbarik2 does not put any restrictions on the representation of the weight distribution. We conducted our experiments on 72 publicly available benchmarks, which have been employed in the evaluation of samplers proposed in the past [13, 21]. The $tilt$ of the benchmarks spans many orders of magnitude, between 1 and $10^{11}$.

**Samplers Tested** The past few years have witnessed a multitude of sampling techniques ranging from variational methods [38], MCMC-based techniques [27, 31], mutation-based sampling [21], importance sampling-based methods [22], knowledge-compilation techniques [25] and the like. The conceptual simplicity of uniform samplers encourages designers to tune their algorithms for uniform sampling, and the standard technique for weighted sampling employs the well-known method of the inverse transform. For the sake of completeness, we provide a detailed discussion of the transformation technique in Appendix C

We perform empirical evaluation with the three state of the art samplers wUniGen, wQuicksampler, and wSTS constructed by augmenting inverse sampling with underlying samplers UniGen [13], Quicksampler [21] and STS(SearchTreeSampler) [22] respectively.

While wUniGen is known to have theoretical guarantees of $\varepsilon$−closeness, there is no theoretical analysis of the distributions generated by wQuicksampler and wSTS. Of the 72 instances, wUniGen can handle only 35 instances while wQuicksampler and wSTS can handle all the 72 instances. The variation in the number of instances that are amenable to sampling for a particular sampler highlights the trade-off between the runtime performance and theoretical guarantees. It is perhaps worth

| Benchmark | *tilt* (maxSamp) | Barbarik2 | | |
|---|---|---|---|---|
| | | wUniGen (samples) | wSTS (samples) | wQuicksampler (samples) |
| s349_3_2 | 28 (3e+07) | A (1e+05) | A (1e+05) | R (22854) |
| s820a_3_2 | 37 (5e+07) | A (96212) | R (87997) | A (2e+05) |
| UserServiceImpl.sk | 140 (6e+08) | A (1e+05) | R (1e+05) | R (4393) |
| LoginService2.sk | 232 (2e+09) | A (1e+05) | R (38044) | R (13350) |
| s349_7_4 | 603 (1e+10) | A (75555) | R (4284) | R (5150) |
| s344_3_2 | 3300 (3e+11) | A (1e+05) | R (59952) | R (5150) |
| s420_new_7_4 | 3549 (4e+11) | A (82312) | A (96659) | R (49955) |
| 54.sk_12_97 | 4e+11 (6e+27) | DNS | R (14012) | R (4627) |
| s641_7_4 | 9e+07 (3e+20) | DNS | R (8747) | A (1e+06) |
| s838_3_2 | 2e+08 (1e+21) | DNS | R (9504) | R (4627) |

Table 1: "A"(resp. "R") represents Barbarik2 returning `ACCEPT`(resp. `REJECT`). maxSamp represents the upper bound on the number of samples required by Barbarik2 to return `ACCEPT/REJECT`.

emphasizing that wQuicksampler and wSTS are significantly more efficient in runtime performance than the ideal sampler WAPS.

**Test Parameters**   We set tolerance parameter $\varepsilon$, intolerance parameter $\eta$, and confidence $\delta$ for Barbarik2 to be and 0.1, 1.6 and 0.2 respectively. The chosen setting of parameters implies that for a given Boolean formula $\varphi$, if the sampler under test $\mathcal{G}(\varphi)$ is $\varepsilon$-close to the ideal sampler, then Barbarik2 returns `ACCEPT` with probability at least 0.8, otherwise if the sampler is $\eta$-far from ideal sampler then Barbarik2 returns `REJECT` with probability at least 0.8. Note that, the number of samples required for `ACCEPT` depends only on the parameters $(\varepsilon, \eta, \delta)$ and $tilt(\mathtt{wt}, \varphi)$. We instantiate Barbarik2Kernel with the values $m = 12$ and $k = 2^m - 1$. Observe that Theorem 1 does not put restrictions on $k$ and $m$.

**Description of the table**   We present the experimental results in Table 1. Due to lack of space, we present results for a subset of benchmarks while the extended table is available in the supplementary material. The first column indicates the name of the benchmark, the second the *tilt*, and the following columns indicate the outcome of the experiments with wUniGen, wSTS and wQuicksampler in that order. Every cell in the table has two entries. In the second column, the first entry shows the value of *tilt* for the corresponding benchmark, while in the other columns, it contains "A" and "R" to indicate the output of Barbarik2 for the corresponding sampler. The second entry for the cells in the column corresponding to *tilt* indicates the theoretical upper bound on the samples required for Barbarik2 to terminate, while for rest of the columns, the second entry indicates the number of samples consumed by Barbarik2 for the corresponding instance and the sampler.

**RQ1**   Our experiments demonstrate that Barbarik2 returns `REJECT` for wQuicksampler on 68 benchmarks and `ACCEPT` on the remaining four benchmarks. For wSTS we found Barbarik2 returned `REJECT` on 62 of the benchmarks and `ACCEPT` on 7 while it times out on the remaining 3. Since wSTS and wQuicksampler are samplers with no formal guarantees and therefore one may expect them to generation distributions away from the ideal distributions. In this context, the results in

Table 1 provide strong evidence for the reasonableness of the *subquery consistency* assumption in practice.

In contrast, Barbarik2 returned `ACCEPT` for wUniGen on all the 35 benchmarks for which wUniGen could sample. Recall, wUniGen formally guarantees $\varepsilon$-closeness of the samples to the required distribution, hence Barbarik2 returning `ACCEPT` on all the benchmarks provides evidence in support of soundness of Barbarik2.

**RQ2**  We also computed the number of samples required by the baseline approach owing to [4]. Since the number of samples is so large that exhaustive experimentation is infeasible, we had to resort to estimating the average time taken by a sampler for a given instance. Based on the estimated time, we can estimate the time taken by the baseline for our benchmark set. We observe that the time taken by the baseline would be over $10^6$ seconds for 43, 42 and 16 benchmarks for wQuicksampler, wSTS and wUniGen respectively. In this context, it is worth highlighting that Barbarik2 terminates within 24 hours for all the instances for all the samplers. We observe that the geometric means of the speedups over the baseline approach are $10^{5.0}, 10^{20.2}$ and 58 for wSTS, wQuicksampler and, wUniGen respectively. The lower speedup in the case of wUniGen owes to its ability to handle only small benchmarks, for which the number of models was not very large. The extended results are available in Appendix D.

**RQ3**  The number of trials required (indicated by the the variable $t$ as on Line 7 of Algorithm 1) depends only on $(\varepsilon, \eta, \delta)$, so for the values we use, $(0.1, 1.6, 0.2)$, we find that we require $t = 14$ trials. The analysis of the algorithm reveals an upper bound on the sample complexity of the tester (See Section 4, Theorem 1) which is quadratic in terms of the $tilt(\mathtt{wt}, \varphi)$. We now return to Table 1 and observe that the number of samples required by Barbarik2 before returning `ACCEPT` were significantly lower than the theoretical bound provided in the second column. Furthermore, as noted earlier, the number of samples required before Barbarik2 returns `REJECT` is typically significantly less than the worst case – a trend demonstrated in Table 1.

## 6   Conclusion

In this paper, we study the problem of verifying whether a probabilistic sampler samples from a given discrete distribution. Existing approaches require samples linear in the size of the sampling set, which is commonly exponentially large. We present a conditional sampling technique that can verify the sampler in sample complexity constant in terms of the sampling set. We also test a prototype implementation of our algorithm against three state-of-the-art samplers.

We noticed that the analytical upper bound on the sample complexity is significantly weak compared to our observed values; this suggests that the bounds could be further tightened. Our algorithm can only deal with those discrete distributions for which the relative probabilities of any two points is easily computable. Since our algorithm does not deal with all possible discrete distributions, extending the approach to other distributions would enable the testing of a broader set of samplers.

## Broader Impact

The recent advances in machine learning techniques have led to increased adoption of the said techniques in safety-critical domains. The usage of a technique in a safety-critical domain necessitates appropriate verification methodology. This paper seeks to take a step in this direction and focused on one core component. Our analysis is probabilistic, and therefore, practical adoption of such techniques requires careful design of frameworks to handle failures.

## Acknowledgements

We are grateful to Teodora Baluta and Arijit Shaw for the technical help and for the useful comments on the earlier drafts of the work. We are grateful to the anonymous reviewers for their constructive feedback that has greatly improved the quality of the paper. This work was supported in part by the National Research Foundation Singapore under its NRF Fellowship Programme [NRF-NRFFAI1-2019-0004] and the AI Singapore Programme [AISG-RP-2018-005], and NUS ODPRT Grant [R-252-

000-685-13]. The computational work for this article was performed on resources of the National Supercomputing Centre, Singapore https://www.nscc.sg. Any opinions, findings and conclusions or recommendations expressed in this material are those of the author(s) and do not reflect the views of National Research Foundation, Singapore.

## Footnotes

*The accompanying tool, available open source, can be found at https://github.com/meelgroup/barbarik. The Appendix is available in the accompanying supplementary material.

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
