[Supplementary Material]

# Appendix

## A Relationship of Barbarik2 with Property Testing

Testing of samplers is basically testing if two distributions $\mathcal{D}_{\mathcal{G}(\varphi)}$ and $\mathcal{A}_{\mathcal{G}(\varphi)}$ are similar, where $\mathcal{G}$ is the sampler under test and $\mathcal{A}$ is the ideal sampler. As stated in the Introduction and the Related Work section, the sub-field of property testing in theoretical computer science has been studying this problem for over two decades and our tester Barbarik2 draws ideas from some of the latest research in this area.

In understanding the closeness between two distributions one may consider a variety of different distance measures. The variation distance (also called the $\ell_1$ distance) is possibly most commonly used. In property testing the problem is to distinguish between the case where the two distributions are $\varepsilon$-close in $\ell_1$ distance from the case where the distributions are $\eta$-far from each other in $\ell_1$ distance. An easier question, called the "equivalence testing of distributions" considers the problem of distinguish identical distributions from distributions that are $\eta$-far from each other in $\ell_1$ distance. The former question, often referred to as the tolerant version of equivalence testing of distributions or estimation of variation distance, is more suitable for various applications. The goal in all the settings is to minimize the sample complexity. The time complexity or other complexity measures are usually not considered in property testing literature.

The problem of equivalence testing of distributions was first considered by [4] and they (along with [37] ) showed that the sample complexity was $\Theta(N^{2/3})$, where $N$ is the size of the support of the distributions. Note that, in the setting of samplers, $N$ is exponential in the input size and hence the number is prohibitively large. The tolerant version of the problem was proved to have even higher sample complexity of $\Theta(N)$ ([37, 36]). This was a significant bottleneck is practicality of these property testing algorithms and the tight lower bounds implied that no improvement was possible for algorithms that has only blackbox access to the distributions. Even the much simpler problem of testing if a distribution is uniform requires $\Omega(\sqrt{N})$ samples.

In [11, 8] a new model for sampling was introduced called the conditional sampling. This model allowed access to the distributions that the standard sampling method (or the blackbox access to the distributions) could not give. It allowed a kind of grey-box access to the distributions. It was shown that in this model only $O(1/\varepsilon^2)$ conditional samples were needed to test if a distribution is uniform or $\varepsilon$-far from uniform. In fact similar conditional sample complexity is sufficient for the non-tolerant version of the equivalence testing of distributions. For the tolerant version of equivalence testing of distributions it was shown that polynomial in $\log(N)$ number of conditional samples suffice. Although this brings down the sample complexity drastically but still it was quite high for practical implementations. On top of that a major obstacle was whether the conditional samples were at all practical and were they implementable.

In [10] they successfully used the idea from the conditional sampling testing to test if samplers are uniform. They crucially used a special kind of conditional sampling. In [8] a concept of pair-conditioning (they called PCOND) was introduced to define a restricted version of the conditional sampling model. A normal conditional sample is obtained by specifying a subset $S$ of the domain of the distribution $\mathcal{D}$ and then drawing a random sample from the conditional distribution $\mathcal{D}|_S$. A PCOND-sample is a normal conditional sample where the subset $S$ is of size 2. In [10] it was shown how this kind of restricted samples can be successfully implemented using a clever use of chain-formulas.

When it come to the more general problems of non-tolerant version of equivalence testing of distributions it can be shown that the sample complexity in the PCOND-model is at least polynomial in $\log N$. The tolerant version has even higher PCOND-sample complexity. Since our primary objective was to have a tester that can be practical and implementable we had to circumvent the problem of high sample complexity and also of implementational issues of conditional sampling. In our tester Barbarik2 we addressed these problems by using another trick from [10], that of, using two different notions of distance - $\ell_\infty$ for closeness and $\ell_1$ for farness. In Barbarik2 we re-designed the sampler and give a proof of correctness in this paradigm using very different techniques as compare to that used in [10].

It is worth noting here that recently conditional sampling and its various variants has been used to design efficient testing and learning algorithms for various other properties of distributions ([1, 28, 5, 9, 17]). Many of these have the potential to be used more efficient and sophisticated testing of samplers and related questions. But the major question is the practicality of the models and the implementability of the algorithms.

## B   Proof of Correctness of Barbarik2

In this section, we present the theoretical analysis of Barbarik2, and the proof of Theorem 1. The proof clearly follows from the the following three lemmas.

**Lemma 2.** *If a sampler $\mathcal{G}$ is $\varepsilon$-close* [3] *to the ideal sampler $\mathcal{A}$ then* Barbarik2 *returns* ACCEPT *with probability at least* $(1 - \delta)$.

**Lemma 3.** *If $\mathcal{G}$ is subquery consistent w.r.t* Barbarik2Kernel *and if the distribution $\mathcal{D}_{\mathcal{G}(\varphi)}$ is $\eta$-far from the ideal sampler then* Barbarik2 *returns* REJECT *with probability at least* $(1 - \delta)$.

**Lemma 4.** *Given $\varepsilon$, $\eta$ and $\delta$,* Barbarik2 *needs at most* $\widetilde{O}\left(\frac{tilt(\mathtt{wt},\varphi)^2}{\eta(\eta-6\varepsilon)^3}\right)$ *samples for any input formula $\varphi$ and weight function* $\mathtt{wt}$*, where the tilde hides a poly logarithmic factor of* $1/\delta, 1/\eta$ *and* $1/(\eta - 6\varepsilon)$.

We will present the proofs of Lemma 2, Lemma 3 and Lemma 4 in Section B.1, Section B.2 and Section B.3 respectively.

In the rest of this section we will use the following notations:

- We use $\mathbb{1}(E)$ to represent the indicator variable for the event $E$.
- We use $R_i$ to denote the event that Barbarik2 returns REJECT in iteration $i$.

For the proof of correctness of our algorithm, we need some standard concentration inequalities. The following versions of the Chernoff Bound will be used.

**Lemma 5.** *Let $Y_1, Y_2, \ldots, Y_n$ be i.i.d 0-1 random variables.*

1. *If $\mathrm{E}[Y_i] \geq \theta \geq 0$, then for any $t \leq \theta$,*

$$\Pr\left[\sum_{j\in[n]}\frac{Y_j}{n} \leq t\right] < exp\left(-\frac{(\theta - t)^2 n}{2\theta}\right)$$

2. *If $\mathrm{E}[Y_i] \leq \theta$, then for any $t \geq \theta$,*

$$\Pr\left[\sum_{j\in[n]}\frac{Y_j}{n} \geq t\right] < exp\left(-\frac{(t - \theta)^2 n}{2t}\right)$$

We are now ready to present the proofs of Lemma 2, Lemma 3 and Lemma 4.

### B.1   Proof of Lemma 2

**Lemma 2.** *If a sampler $\mathcal{G}$ is $\varepsilon$-close* [4] *to the ideal sampler $\mathcal{A}$ then* Barbarik2 *returns* ACCEPT *with probability at least* $(1 - \delta)$.

For the proof of Lemma 2 we will firstly show (in Lemma 6) that in each iteration of the loop, the probability that Barbarik2 returns REJECT is less than $\delta/t$ and then the proof of Lemma 2 follows by the application of the Chernoff Bound. Recall that $R_i$ denotes the event that Barbarik2 returns REJECT in iteration $i$.

**Lemma 6.** *If sampler $\mathcal{G}$ is $\varepsilon$-close to an ideal sampler $\mathcal{A}$ then the probability that* Barbarik2 *returns* REJECT *in any particular iteration of the loop, is atmost $\delta/t$. Then*

$$\Pr\left[\overline{R_i} \mid \bigwedge_{j\in[i-1]} \overline{R_j}\right] \geq \left(1 - \frac{\delta}{t}\right)$$

*Proof.* (of Lemma 6) Barbarik2 returns REJECT in the $i$th iteration if the $Bias$ (in the $i$th iteration) is more than $T$, where $T = \frac{L+H}{2}$ with

$$L = \frac{(1+\varepsilon)p_{\mathcal{A}}(\varphi, S, \sigma_1)}{(1+\varepsilon)p_{\mathcal{A}}(\varphi, S, \sigma_1) + (1-\varepsilon)p_{\mathcal{A}}(\varphi, S, \sigma_2)}$$

And since, by definition, all the elements in $\Gamma_1$, $\Gamma_2$ and $\Gamma_3$ are obtained by drawing independent samples from $\mathcal{D}_{\mathcal{G}(\varphi)}$, $\mathcal{D}_{\mathcal{A}(\varphi)}$ and $\mathcal{D}_{\mathcal{G}(\hat{\varphi})}$ respectively so

$$\Pr\left[\overline{R_i} \mid \bigwedge_{j\in[i-1]} \overline{R_j}\right] = \Pr\left[\, Bias \leq T \text{ in the } i\text{th iteration}\right]$$

$$= 1 - \Pr\left[\, Bias > T \text{ in the } i\text{th iteration}\right]$$

$$= 1 - \Pr\left[\sum_{j\in[N]} \frac{\mathbb{1}(\Gamma_3[j]_{\downarrow S} = \sigma_1)}{N} > T\right]$$

Note that the random variables $\mathbb{1}(\Gamma_3[j]_{\downarrow S} = \sigma_1)$ are an i.i.d 0-1 random variable. And since the sampler $\mathcal{G}$ is assumed to be $\varepsilon$-close to the ideal sampler so we have

$$(1-\varepsilon)p_{\mathcal{A}}(\hat{\varphi}, \Gamma_3[j]) \leq p_{\mathcal{G}}(\hat{\varphi}, \Gamma_3[j]) \leq (1+\varepsilon)p_{\mathcal{A}}(\hat{\varphi}, \Gamma_3[j]).$$

Thus we have,

$$\mathrm{E}[\mathbb{1}(\Gamma_3[j]_{\downarrow S} = \sigma_1)] = p_{\mathcal{G}}(\hat{\varphi}, S, \sigma_1) \leq (1+\varepsilon)p_{\mathcal{A}}(\hat{\varphi}, S, \sigma_1)$$

Now, since $p_{\mathcal{A}}(\hat{\varphi}, S, \sigma_1) = \frac{p_{\mathcal{A}}(\varphi, S, \sigma_1)}{p_{\mathcal{A}}(\varphi, S, \sigma_1) + p_{\mathcal{A}}(\varphi, S, \sigma_2)}$ we have

$$\mathrm{E}[\mathbb{1}(\Gamma_3[j]_{\downarrow S} = \sigma_1)] = p_{\mathcal{G}}(\hat{\varphi}, S, \sigma_1) \leq \frac{(1+\varepsilon)p_{\mathcal{A}}(\varphi, S, \sigma_1)}{p_{\mathcal{A}}(\varphi, S, \sigma_1) + p_{\mathcal{A}}(\varphi, S, \sigma_2)} \qquad (1)$$

Similarly, we have that

$$\mathrm{E}[\mathbb{1}(\Gamma_3[j]_{\downarrow S} = \sigma_2)] = p_{\mathcal{G}}(\hat{\varphi}, S, \sigma_2) \geq \frac{(1-\varepsilon)p_{\mathcal{A}}(\varphi, S, \sigma_2)}{p_{\mathcal{A}}(\varphi, S, \sigma_1) + p_{\mathcal{A}}(\varphi, S, \sigma_2)} \qquad (2)$$

Now we consider two cases depending on whether $p_{\mathcal{A}}(\varphi, S, \sigma_1)$ is greater or lesser than $p_{\mathcal{A}}(\varphi, S, \sigma_2)$. If $p_{\mathcal{A}}(\varphi, S, \sigma_1) \leq p_{\mathcal{A}}(\varphi, S, \sigma_2)$ then from Equation 1 we have

$$\mathrm{E}[\mathbb{1}(\Gamma_3[j]_{\downarrow S} = \sigma_1)] = p_{\mathcal{A}}(\hat{\varphi}, S, \sigma_1)$$

$$\leq \frac{(1+\varepsilon)p_{\mathcal{A}}(\varphi, S, \sigma_1)}{p_{\mathcal{A}}(\varphi, S, \sigma_1) + p_{\mathcal{A}}(\varphi, S, \sigma_2)}$$

$$\leq \frac{(1+\varepsilon)p_{\mathcal{A}}(\varphi, S, \sigma_1)}{(1+\varepsilon)p_{\mathcal{A}}(\varphi, S, \sigma_1) + (1-\varepsilon)p_{\mathcal{A}}(\varphi, S, \sigma_2)} = L \qquad (3)$$

But if $p_{\mathcal{A}}(\varphi, S, \sigma_1) \geq p_{\mathcal{A}}(\varphi, S, \sigma_2)$ then from Equation 1 we have

$$\mathrm{E}[\mathbb{1}(\Gamma_3[j]_{\downarrow S} = \sigma_2)] = p_{\mathcal{A}}(\hat{\varphi}, S, \sigma_2)$$

$$\geq \frac{(1-\varepsilon)p_{\mathcal{A}}(\varphi, S, \sigma_2)}{p_{\mathcal{A}}(\varphi, S, \sigma_1) + p_{\mathcal{A}}(\varphi, S, \sigma_2)}$$

$$\geq \frac{(1-\varepsilon)p_{\mathcal{A}}(\varphi, S, \sigma_2)}{(1+\varepsilon)p_{\mathcal{A}}(\varphi, S, \sigma_1) + (1-\varepsilon)p_{\mathcal{A}}(\varphi, S, \sigma_2)}$$

And in that case since $p_{\mathcal{A}}(\hat{\varphi}, S, \sigma_1) + p_{\mathcal{A}}(\hat{\varphi}, S, \sigma_2) = 1$ we have

$$
\begin{aligned}
\mathrm{E}[\mathbb{1}(\Gamma_3[j]_{\downarrow S} = \sigma_1)] &= p_{\mathcal{A}}(\hat{\varphi}, S, \sigma_1) \\
&= 1 - p_{\mathcal{A}}(\hat{\varphi}, S, \sigma_2) \\
&\leq 1 - \left( \frac{(1-\varepsilon)p_{\mathcal{A}}(\varphi, S, \sigma_2)}{(1+\varepsilon)p_{\mathcal{A}}(\varphi, S, \sigma_1) + (1-\varepsilon)p_{\mathcal{A}}(\varphi, S, \sigma_2)} \right) \\
&\leq \frac{(1+\varepsilon)p_{\mathcal{A}}(\varphi, S, \sigma_1)}{(1+\varepsilon)p_{\mathcal{A}}(\varphi, S, \sigma_1) + (1-\varepsilon)p_{\mathcal{A}}(\varphi, S, \sigma_2)} = L
\end{aligned}
\tag{4}
$$

Thus in either case, from Equation (3) and Equation (4) we have $\mathrm{E}[\mathbb{1}(\Gamma_3[j]_{\downarrow S} = \sigma_1)] \leq L$. Now applying the Chernoff bound from Lemma 5 we have

$$
\begin{aligned}
\Pr[Bias \geq T] = \Pr\left[ \sum_{j \in [N]} \frac{\mathbb{1}(\Gamma_3[j]_{\downarrow S} = \sigma_1)}{N} > T \right] \\
= exp\left( -\frac{(T-L)^2 N}{2L} \right) = exp\left( -\frac{(H-L)^2 N}{8L} \right) \\
\leq exp\left( -\frac{(H-L)^2 N}{8H} \right) \quad \text{Because } [H \geq L] \\
\leq \frac{\delta}{t},
\end{aligned}
\tag{5}
\tag{6}
$$

where the inequality in line (5) follows because $H \geq L$ when $\varepsilon \leq 1/3$ and $\eta \geq 6\varepsilon$[5] and last inequality follows because $N = n.H/(H-L)^2$ where $n = 8\log(t/\delta)$. $\qquad\square$

*Proof.* (of Lemma 2) Let $R_i$ denote the event that Barbarik2 returns REJECT in iteration $i$ and $\overline{R}$ denote the event that Barbarik2 returns ACCEPT. Thus $\overline{R} = \cap_i \overline{R_i}$.

In the $i^{th}$ iteration if the bias is less than the threshold, Barbarik2 fails to REJECT. Thus from Lemma 6 if the sampler $\mathcal{G}$ is $\varepsilon$-close to the ideal sampler $\mathcal{A}$ then

$$
\Pr\left[ \overline{R_i} | \bigwedge_{j \in [i-1]} \overline{R_j} \right] \geq 1 - \frac{\delta}{t}
$$

If Barbarik2 has not returned REJECT in any of the iteration then after the last iteration Barbarik2 returns ACCEPT. The probability of Barbarik2 returning ACCEPT (event $\overline{R}$) is

$$
\Pr\left[ \overline{R} \right] \geq \prod_{i \in [t]} \Pr\left[ \overline{R_i} | \bigwedge_{j \in [i-1]} \overline{R_j} \right] \geq \left( 1 - \frac{\delta}{t} \right)^t \geq 1 - \delta
$$

$\qquad\square$

## B.2 Proof of Lemma 3

**Lemma 3.** *If $\mathcal{G}$ is subquery consistent w.r.t* Barbarik2Kernel *and if the distribution $\mathcal{D}_{\mathcal{G}(\varphi)}$ is $\eta$-far from the ideal sampler then* Barbarik2 *returns* REJECT *with probability at least $(1-\delta)$.*

*Proof.* To prove the Lemma, we will start by splitting the set $R_\varphi$ into disjoint subsets depending on the distribution $D_{\mathcal{G}(\varphi)}$.

**Definition 10.** *We define the following sets for use in the soundness proof:*

- $D = \{x \in R_\varphi : \ p_{\mathcal{G}}(\varphi, x) \leq p_{\mathcal{A}}(\varphi, x)\}$

- $U = R_\varphi \setminus D$

- $U_0 = \{x \in R_\varphi : \ p_\mathcal{A}(\varphi, x) < p_\mathcal{G}(\varphi, x) \le \left(1 + \frac{\eta + 6\varepsilon}{4}\right) p_\mathcal{A}(\varphi, x)\}$.

- $U_1 = \{x \in R_\varphi : \ \left(1 + \frac{\eta + 6\varepsilon}{4}\right) p_\mathcal{A}(\varphi, x) < p_\mathcal{G}(\varphi, x)\}$

Recall, $R_i$ is the event that Barbarik2 returns REJECT in the $i$th iteration of the for loop. Then the following lemmas helps us to lower bound the probability of $\Gamma_1[i] \in U_1 \wedge \Gamma_2[i] \in D$ and the probability of $R_i$ under the condition that $\Gamma_1[i] \in U_1 \wedge \Gamma_2[i] \in D$.

**Lemma 7.** *If the sampler $\mathcal{G}$ is $\eta$-far from the ideal sampler then*

$$\Pr\left[R_i | (\bigwedge_{j \in [i-1]} \overline{R_j}) \wedge (\Gamma_1[i] \in U_1 \wedge \Gamma_2[i] \in D)\right] \ge \frac{4}{5}.$$

**Lemma 8.** *If the sampler $\mathcal{G}$ is $\eta$-far from the ideal sampler on input $\varphi$ then*

$$\Pr\left[\Gamma_1[i] \in U_1 \wedge \Gamma_2[i] \in D\right] \ge \frac{\eta(\eta - 6\varepsilon)}{8}.$$

And now using Lemmas 8 and 7 we can complete the proof of soundness. The probability that Barbarik2 returns REJECT in the $i$th iteration of the for loop is

$$
\begin{aligned}
&\Pr\left[R_i \mid \bigwedge_{j \in [i-1]} \overline{R_j}\right] \\
&= \Pr\left[R_i \mid (\bigwedge_{j \in [i-1]} \overline{R_j}) \wedge (\Gamma_1[i] \in U_1 \wedge \Gamma_2[i] \in D)\right] \cdot \Pr[\Gamma_1[i] \in U_1 \wedge \Gamma_2[i] \in D] \\
&\ge \left(\frac{4}{5}\right) \frac{\eta(\eta - 6\varepsilon)}{8} \quad \text{[From Lemma 8 and Lemma 7]} \quad (7)
\end{aligned}
$$

The probability of Barbarik2 returning REJECT in any iteration (event $R$) is given by

$$
\begin{aligned}
\Pr\left[\cup_i R_i\right] &= 1 - \prod_{i \in [t]} \Pr\left[\overline{R_i} \mid \bigwedge_{j \in [i-1]} \overline{R_j}\right] \\
&\ge 1 - \prod_{i \in [t]} \left(1 - \frac{\eta(\eta - 6\varepsilon)}{10}\right) \quad \text{[Using Equation (7)]} \\
&\ge 1 - \left(1 - \frac{\eta(\eta - 6\varepsilon)}{10}\right)^t
\end{aligned}
$$

Substituting $t$, $\quad \ge 1 - \delta$

$\square$

Now to complete the proof of Lemma 3 we have to prove the Lemma 7 and Lemma 8. They are presented next.

**Lemma 7.** *If the sampler $\mathcal{G}$ is $\eta$-far from the ideal sampler then*

$$\Pr\left[R_i | (\bigwedge_{j \in [i-1]} \overline{R_j}) \wedge (\Gamma_1[i] \in U_1 \wedge \Gamma_2[i] \in D)\right] \ge \frac{4}{5}.$$

*Proof.* (of Lemma 7) Let us assume $\Gamma_1[i] \in U_1$ and $\Gamma_2[i] \in D$. That is, we have $p_{\mathcal{G}}(\varphi, S, \Gamma_2[i]) \leq p_{\mathcal{A}}(\varphi, S, \Gamma_2[i])$ and $p_{\mathcal{G}}(\varphi, S, \Gamma_1[i]) > \left(1 + \frac{\eta + 6\varepsilon}{4}\right) p_{\mathcal{A}}(\varphi, S, \Gamma_1[i])$. It follows that

$$\frac{p_{\mathcal{G}}(\varphi, S, \Gamma_1[i])}{p_{\mathcal{G}}(\varphi, S, \Gamma_2[i])} \geq \left(1 + \frac{6\varepsilon + \eta}{4}\right) \cdot \frac{p_{\mathcal{A}}(\varphi, S, \Gamma_1[i])}{p_{\mathcal{A}}(\varphi, S, \Gamma_2[i])} \tag{8}$$

Since $\forall x > 0, a/b > x \implies a/(a+b) > x/(x+1)$, we have from Equation 8

$$\frac{p_{\mathcal{G}}(\varphi, S, \Gamma_1[i])}{p_{\mathcal{G}}(\varphi, S, \Gamma_2[i]) + p_{\mathcal{G}}(\varphi, S, \Gamma_1[i])}$$
$$\geq \left(1 + \frac{6\varepsilon + \eta}{4}\right) \cdot \frac{p_{\mathcal{A}}(\varphi, S, \Gamma_1[i])}{p_{\mathcal{A}}(\varphi, S, \Gamma_2[i])} \cdot \left(1 + \left(1 + \frac{6\varepsilon + \eta}{4}\right) \cdot \frac{p_{\mathcal{A}}(\varphi, S, \Gamma_1[i])}{p_{\mathcal{A}}(\varphi, S, \Gamma_2[i])}\right)^{-1}$$

Thus we have

$$\mathrm{E}[\mathbb{1}(\Gamma_3[j]_{\downarrow S} = \sigma_1)] = p_{\mathcal{G}}(\hat{\sigma}, S, \Gamma_1[i])$$
$$= \frac{p_{\mathcal{G}}(\varphi, S, \Gamma_1[i])}{p_{\mathcal{G}}(\varphi, S, \Gamma_2[i]) + p_{\mathcal{G}}(\varphi, S, \Gamma_1[i])} \quad [\text{ by the subquery consistent sampler assumption}]$$
$$\geq \left(1 + \frac{6\varepsilon + \eta}{4}\right) \cdot \frac{p_{\mathcal{A}}(\varphi, S, \Gamma_1[i])}{p_{\mathcal{A}}(\varphi, S, \Gamma_2[i])} \cdot \left(1 + \left(1 + \frac{6\varepsilon + \eta}{4}\right) \cdot \frac{p_{\mathcal{A}}(\varphi, S, \Gamma_1[i])}{p_{\mathcal{A}}(\varphi, S, \Gamma_2[i])}\right)^{-1}$$
$$= H \quad [\text{By definition of } H] \tag{9}$$

Barbarik2 returns REJECT in the $i$th iteration if the $Bias$ (in the $i$th iteration) is more than $T$, where $T = \frac{L+H}{2}$ with

$$H = \frac{(1 + \frac{6\varepsilon + \eta}{4}) p_{\mathcal{A}}(\varphi, S, \sigma_1)}{(1 + \frac{6\varepsilon + \eta}{4}) p_{\mathcal{A}}(\varphi, S, \sigma_1) + p_{\mathcal{A}}(\varphi, S, \sigma_2)}$$

And since, by definition, all the elements in $\Gamma_1$, $\Gamma_2$ and $\Gamma_3$ are obtained by drawing independent samples from $\mathcal{D}_{\mathcal{G}(\varphi)}$, $\mathcal{D}_{\mathcal{A}(\varphi)}$ and $\mathcal{D}_{\mathcal{G}(\hat{\varphi})}$ respectively so

$$\Pr\left[R_i \mid (\bigwedge_{j \in [i-1]} \overline{R_j}) \bigwedge (\Gamma_1[i] \in U_1 \wedge \Gamma_2[i] \in D)\right]$$
$$= \Pr[\, Bias > T \text{ in the } i\text{th iteration} \mid (\Gamma_1[i] \in U_1 \wedge \Gamma_2[i] \in D)]$$
$$= \Pr\left[\sum_{j \in [N]} \frac{\mathbb{1}(\Gamma_3[j]_{\downarrow S} = \sigma_1)}{N} \geq T \mid (\Gamma_1[i] \in U_1 \wedge \Gamma_2[i] \in D)\right]$$

Now since $\mathbb{1}(\Gamma_3[j]_{\downarrow S} = \sigma_1)$ are i.i.d 0-1 random variables and since $\Gamma_1[i] \in U_1$ and $\Gamma_2[i] \in D$ implies $\mathrm{E}[\mathbb{1}(\Gamma_3[j]_{\downarrow S} = \sigma_1)] \geq H$ (from Equation 9) by applying Chernoff bound from Lemma 5 we have:

$$\Pr\left[\frac{1}{N} \sum_{j \in [N]} \mathbb{1}(\Gamma_3[j]_{\downarrow S} = \sigma_1) \geq T\right] \leq exp\left(-\frac{(H-T)^2 N}{8H}\right)$$
$$\text{by the choice of N} \quad \leq \frac{\delta}{t}$$
$$\text{since } \delta < 0.5 \text{ and } t \geq 3 \quad \leq 1/5$$

$\square$

**Lemma 8.** *If the sampler $\mathcal{G}$ is $\eta$-far from the ideal sampler on input $\varphi$ then*

$$\Pr[\Gamma_1[i] \in U_1 \wedge \Gamma_2[i] \in D] \geq \frac{\eta(\eta - 6\varepsilon)}{8}.$$

*Proof.* of Lemma 8) Since the sampler $\mathcal{G}$ is $\varepsilon$-far from the ideal sampler on input $\varphi$ so, the $\ell_1$ distance between $\mathcal{D}_{\mathcal{G}}(\varphi)$ and $\mathcal{D}_{\mathcal{A}}(\varphi)$ is at least $\eta$. By the definition of sets $U$ and $D$ we have,

$$\sum_{x \in U}(p_{\mathcal{G}}(\varphi, x) - p_{\mathcal{A}}(\varphi, x)) = \sum_{x \in D}(p_{\mathcal{A}}(\varphi, x) - p_{\mathcal{G}}(\varphi, x)) \geq \frac{\eta}{2} \tag{10}$$

Now by definition of $U_0$, we have

$$\sum_{x \in U_0} (p_{\mathcal{G}}(\varphi, x) - p_{\mathcal{A}}(\varphi, x)) < \frac{\eta + 6\varepsilon}{4} \quad \sum_{x \in U_0} p_{\mathcal{A}}(\varphi, x) < \frac{\eta + 6\varepsilon}{4} \tag{11}$$

As $U = U_0 \cup U_1$,

$$\sum_{x \in U_1} (p_{\mathcal{G}}(\varphi, x) - p_{\mathcal{A}}(\varphi, x))$$
$$= \sum_{x \in U} (p_{\mathcal{G}}(\varphi, x) - p_{\mathcal{A}}(\varphi, x)) - \sum_{x \in U_0} (p_{\mathcal{G}}(\varphi, x) - p_{\mathcal{A}}(\varphi, x)) \tag{12}$$

Substituting Equation (11) and Equation (10) in Equation (12) we get:-

$$\sum_{x \in U_1} (p_{\mathcal{G}}(\varphi, x) - p_{\mathcal{A}}(\varphi, x)) \geq \frac{\eta}{2} - \frac{\eta + 6\varepsilon}{4} = \frac{\eta - 6\varepsilon}{4}$$

$$\text{Therefore,} \sum_{x \in U_1} p_{\mathcal{G}}(\varphi, x) \geq \frac{\eta - 6\varepsilon}{4}$$

Thus we have,

$$\Pr[\Gamma_1[i] \in U_1] = \sum_{x \in U_1} p_{\mathcal{G}}(\varphi, x) \geq \frac{\eta - 6\varepsilon}{4} \tag{13}$$

From Equation (10) we know that,

$$\Pr[\Gamma_2[i] \in D] = \sum_{x \in D} p_{\mathcal{A}}(\varphi, x) \geq \frac{\eta}{2} \tag{14}$$

Since $\Gamma_1[i] \in U_1$ and $\Gamma_2[i] \in D$ are independent events, putting together Equation (13) and Equation (14), we see that

$$\Pr[\Gamma_1[i] \in U_1 \wedge \Gamma_2[i] \in D] \geq \frac{\eta(\eta - 6\varepsilon)}{8}$$

$\square$

### B.3 Proof of Lemma 4

**Lemma 4.** *Given $\varepsilon$, $\eta$ and $\delta$,* Barbarik2 *needs at most $\widetilde{O}\left(\frac{tilt(\mathtt{wt}, \varphi)^2}{\eta(\eta - 6\varepsilon)^3}\right)$ samples for any input formula $\varphi$ and weight function* wt*, where the tilde hides a poly logarithmic factor of $1/\delta, 1/\eta$ and $1/(\eta - 6\varepsilon)$.*

*Proof.* From Algorithm 1, line 1, we see that the number of trials is:

$$t = \frac{ln(1/\delta)}{ln\left(\frac{10}{10 - \eta(\eta - 6\varepsilon)}\right)}$$

$$(ln(x) \leq x - 1) \quad t \leq ln(1/\delta) \frac{10}{(\eta(\eta - 6\varepsilon))}$$

In every iteration we calculate a value $N$ according to the expression:

$$N = 8ln\left(\frac{t}{\delta}\right) \cdot \frac{\alpha \cdot hi}{1 + \alpha \cdot hi} \cdot \left(\frac{\alpha \cdot hi}{1 + \alpha \cdot hi} - \frac{\alpha \cdot lo}{1 + \alpha \cdot lo}\right)^{-2}$$

$$= 8ln\left(\frac{t}{\delta}\right) \cdot \left(\frac{1}{hi - lo}\right)^2 \cdot hi \cdot \frac{1 + \alpha \cdot hi}{\alpha} \cdot (1 + \alpha \cdot lo)^2$$

$$(1 < lo < hi < 2) \quad < 8ln\left(\frac{t}{\delta}\right) \cdot \left(\frac{1}{hi - lo}\right)^2 \cdot 2 \cdot \frac{1 + \alpha \cdot 2}{\alpha} \cdot (1 + \alpha \cdot 2)^2$$

On Line (11) in Algorithm 1 we define:

$$\alpha = \frac{\mathtt{wt}(\sigma_1)}{\mathtt{wt}(\sigma_2)}$$

$$\text{(Definition 4)} \quad tilt(\mathtt{wt}, \varphi) = \max_{\sigma_1, \sigma_2 \in R_\varphi} \frac{\mathtt{wt}(\sigma_1)}{\mathtt{wt}(\sigma_2)}$$

Thus, $\alpha \leq tilt(\mathtt{wt}, \varphi)$. Substituting the values of $\alpha, lo$ and $hi$, we get:

$$N < 8ln\left(\frac{t}{\delta}\right) \cdot \left(\frac{tilt(\mathtt{wt}, \varphi)}{\eta - 6\varepsilon}\right)^2$$

The maximum number of samples drawn after $t$ trials is:

$$2t + tN < 2tN$$

$$\text{(Substituting for t,N)} \quad < 8ln\left(\frac{1}{\delta} \cdot \frac{10 \cdot ln(1/\delta)}{\eta(\eta - 6\varepsilon)}\right) \times \frac{10 \cdot ln(1/\delta)}{\eta(\eta - 6\varepsilon)} \times \frac{tilt(\mathtt{wt}, \varphi)^2}{(\eta - 6\varepsilon)^2}$$

$$= \tilde{O}\left(\frac{tilt(\mathtt{wt}, \varphi)^2}{\eta(\eta - 6\varepsilon)^3}\right)$$

$\square$

## C   Log-Linear Distributions and Inverse Transform Sampling

Log-linear models capture wide class of distributions of interest including those arising from graphical models, conditional random fields, skip-gram models [34]. Formally, for $\sigma \in \{0, 1\}^n$, we define

$$\Pr[\sigma|\theta] \propto e^{\theta \cdot \sigma}$$

Following Chavira and Darwiche [16], we describe the following equivalent representation, called literal-weighted functions, of log-linear models.

**Definition 11** (Literal-Weighted Functions). *For a CNF formula $\varphi$ and set $S \subseteq Supp(\varphi)$, a weight function $\mathtt{wt} : \{0, 1\}^{|S|} \to (0, 1)$ is called a literal-weighted function if there is a map $\mathtt{W} : S \to (0, 1)$ such that for any assignment $\sigma \in R_{\varphi \downarrow S}$*

$$\mathtt{wt}(\sigma) = \prod_{x \in \sigma} \begin{cases} \mathtt{W}(x) & if \quad x = 1 \\ 1 - \mathtt{W}(x) & if \quad x = 0 \end{cases}$$

*In this case we call $\mathtt{wt}$ a literal-weighted function w.r.t. $\mathtt{W}$. And note that we have $\Pr[\sigma] \propto \mathtt{wt}(\sigma)$.*

We now discuss the standard technique of inverse transform sampling for completeness. For completeness, we follow the description due to Chakrborty et al [15].

**Lemma 9.** *For any $\varepsilon$-close uniform sampler $\mathcal{V}$, any CNF formula $\varphi$ with support $S$ and a literal-weighted function $\mathtt{wt} : \{0, 1\}^{|S|} \to (0, 1)$, we can construct a $\hat{\varphi}$ s.t.*

$$\forall_{\sigma \in R_\varphi}, \frac{(1 - \varepsilon)\mathtt{wt}(\sigma)}{\sum_{\sigma' \in R_\varphi} \mathtt{wt}(\sigma')} \leq p_\mathcal{V}(\hat{\varphi}, S, \sigma) \leq \frac{(1 + \varepsilon)\mathtt{wt}(\sigma)}{\sum_{\sigma' \in R_\varphi} \mathtt{wt}(\sigma')}$$

*Proof.* Let $S_i = \{x_{i,1}, \cdots, x_{i,m_i}\}$ be a set of $m_i$ "fresh" variables (i.e. variables that were not used before) for each $x_i \in S$. Given any integer $m_i > 0$ and a positive odd number $k_i < 2^{m_i}$, we construct $\varphi_{k_i, m_i}(x_{i,1}, \cdots x_{i,m_i})$ using the chain formula construction in [15] such that $|R_{\varphi_{k_i, m_i}}| = k$. For notational clarity, we simply write $\varphi_{k_i, m_i}$ when the arguments of the chain formula are clear from context. For each variable $x_i \in S$, such that $\mathtt{W}(x_i^1) = \frac{k_i}{2^{m_i}}$, and $\mathtt{W}(x_i^0) = 1 - \mathtt{W}(x_i)$, let $(x_i \leftrightarrow \varphi_{k_i, m_i})$ be the representative clause. Thus let $\varphi^{CNF} = \bigwedge_{i \in S}(x_i \leftrightarrow \varphi_{k_i, m_i})$. We then define the formula $\hat{\varphi}$ as follows:

$$\hat{\varphi} = \varphi \wedge \varphi^{CNF}$$

We can see that model count of the formula $|R_{\hat{\varphi}}|$ can be given by:

$$|R_{\hat{\varphi}}| = \sum_{\hat{\sigma} \in R_{\hat{\varphi}}} 1 = \sum_{\sigma \in R_{\varphi}} \sum_{(\hat{\sigma} \in R_{\hat{\varphi}} : \hat{\sigma}_{\downarrow S} = \sigma)} 1 \qquad (15)$$

Since the representative formula of every variable uses a fresh set of variables, we have from the structure of $\hat{\varphi}$ that if $\sigma$ is a witness of $\varphi$ then:

$$\sum_{(\hat{\sigma} \in R_{\hat{\varphi}} : \hat{\sigma}_{\downarrow S} = \sigma)} 1 = \prod_{i \in \sigma^0} (2^{m_i} - k_i) \prod_{i \in \sigma^1} k_i \qquad (16)$$

For any $\sigma \in R_{\varphi}$:

$$
\begin{aligned}
p_{\mathcal{U}}(\hat{\varphi}, S, \sigma) &= \sum_{(\hat{\sigma} \in R_{\hat{\varphi}} : \hat{\sigma}_{\downarrow S} = \sigma)} p_{\mathcal{U}}(\hat{\varphi}, \hat{S}, \hat{\sigma}) \\
&= \sum_{(\hat{\sigma} \in R_{\hat{\varphi}} : \hat{\sigma}_{\downarrow S} = \sigma)} \frac{1}{|R_{\hat{\varphi}}|} \\
&= \frac{\sum_{(\hat{\sigma} \in R_{\hat{\varphi}} : \hat{\sigma}_{\downarrow S} = \sigma)} 1}{\sum_{\sigma' \in R_{\varphi}} \sum_{(\hat{\sigma} \in R_{\hat{\varphi}} : \hat{\sigma}_{\downarrow S} = \sigma')} 1} \quad \text{Using (15)} \\
&= \frac{\prod_{i \in \sigma^0} (2^{m_i} - k_i) \prod_{i \in \sigma^1} k_i}{\sum_{\sigma' \in R_{\varphi}} \prod_{i \in \sigma'^0} (2^{m_i} - k_i) \prod_{i \in \sigma'^1} k_i} \quad \text{Using (16)} \\
&= \frac{\prod_{i \in \sigma^0} (2^{m_i} - k_i) \prod_{i \in \sigma^1} k_i}{\prod_{i \in S} 2^{m_i}} \cdot \frac{\prod_{i \in S} 2^{m_i}}{\sum_{\sigma' \in R_{\varphi}} \prod_{i \in \sigma'^0} (2^{m_i} - k_i) \prod_{i \in \sigma'^1} k_i} \\
&= \frac{\prod_{i \in S} \mathtt{W}(\sigma_{\downarrow x_i})}{\sum_{\sigma' \in R_{\varphi}} \prod_{i \in S} \mathtt{W}(\sigma'_{\downarrow x_i})} \\
&= \frac{\mathtt{wt}(\sigma)}{\sum_{\sigma' \in R_{\varphi}} \mathtt{wt}(\sigma')} \qquad (17)
\end{aligned}
$$

From the definition of $\varepsilon$-additive closeness (Def. 5) we have:

$$(1 - \varepsilon) p_{\mathcal{U}}(\varphi, S, \sigma) \leq p_{\mathcal{V}}(\varphi, S, \sigma) \leq (1 + \varepsilon) p_{\mathcal{U}}(\varphi, S, \sigma)$$

Substituing into 17, we get:

$$\forall_{\sigma \in R_{\varphi}}, \frac{(1 - \varepsilon) \mathtt{wt}(\sigma)}{\sum_{\sigma' \in R_{\varphi}} \mathtt{wt}(\sigma')} \leq p_{\mathcal{V}}(\hat{\varphi}, S, \sigma) \leq \frac{(1 + \varepsilon) \mathtt{wt}(\sigma)}{\sum_{\sigma' \in R_{\varphi}} \mathtt{wt}(\sigma')}$$

$\square$

*Remark* 1. It is worth noting that Lemma 9 implies that if $\mathcal{V}$ is $\varepsilon$-close uniform sampler $\mathcal{V}$ then it can be used as a blackbox to obtain a $\varepsilon$-close to an ideal sampler w.r.t any literal-weighted function wt. It should also be noted that Lemma 9 does not imply that if $\mathcal{V}$ is $\eta$-far from a uniform sampler, then the new sampler (obtained using the above transformation) is also far from the ideal sampler w.r.t wt. Therefore, to test whether $p_{\mathcal{V}}(\hat{\varphi}, S, \sigma)$ is close to ideal sampler, one can not rely on merely testing uniformity of $\mathcal{V}$.

# D    Extended Tables of Results

## D.1    Comparing sample complexity.

"A"("R") represent Barbarik2 returning `ACCEPT(REJECT)`. "DNS" is used against those instances on which the indicated sampler Did Not Sample. "-" indicates that Barbarik2 timed out on that particular instance on the indicated sampler. Note that "DNS" is different from "-" as "DNS" indicates the failure of the underlying sampler to sample the initial set of samples, while "-" indicates the failure of Barbarik2 to finish within the timeout period. The timeout was set to 50,000 seconds for wSTS and wQuicksampler, while for wUniGen it was 24 hours.

Table 2: The Extended Table

| Benchmark | $tilt$ (maxSamp) | Barbarik2 | | |
|---|---|---|---|---|
| | | wUniGen (samples) | wSTS (samples) | wQuicksampler (samples) |
| 107.sk_3_90 | 1 (2e+05) | DNS | R (5146) | R (6009) |
| tableBasedAddition.sk | 1 (2e+05) | DNS | R (6009) | R (24534) |
| 55.sk_3_46 | 1 (2e+05) | DNS | R (8911) | R (4354) |
| 111.sk_2_36 | 1 (2e+05) | DNS | R (23543) | R (5150) |
| 17.sk_3_45 | 1 (2e+05) | DNS | R (1e+05) | R (4677) |
| 80.sk_2_48 | 1 (2e+05) | DNS | R (4284) | R (4627) |
| 27.sk_3_32 | 1 (2e+05) | A (1e+05) | R (25329) | R (6009) |
| 70.sk_3_40 | 1 (2e+05) | DNS | R (10402) | R (17704) |
| 32.sk_4_38 | 1 (2e+05) | A (1e+05) | R (18081) | R (14682) |
| 84.sk_4_77 | 1 (2e+05) | DNS | R (5146) | R (4354) |
| 53.sk_4_32 | 1 (2e+05) | A (1e+05) | R (35618) | R (6009) |
| s35932_3_2 | 3 (6e+05) | DNS | TO | R (11756) |
| s35932_7_4 | 3 (6e+05) | DNS | TO | R (11756) |
| s832a_3_2 | 3 (6e+05) | A (1e+05) | R (8708) | R (54138) |
| 109.sk_4_36 | 8 (3e+06) | DNS | R (26218) | R (6009) |
| 77.sk_3_44 | 11 (5e+06) | DNS | R (47582) | R (47907) |
| s35932_15_7 | 12 (6e+06) | DNS | TO | R (4354) |
| s832a_7_4 | 15 (8e+06) | A (1e+05) | R (4393) | R (13350) |
| 51.sk_4_38 | 18 (1e+07) | A (78661) | R (4284) | R (4627) |
| 29.sk_3_45 | 26 (2e+07) | DNS | R (4284) | R (55989) |
| 81.sk_5_51 | 27 (3e+07) | DNS | R (28409) | A (2e+05) |
| s349_3_2 | 28 (3e+07) | A (1e+05) | A (1e+05) | R (22854) |
| s298_3_2 | 32 (3e+07) | A (1e+05) | R (80883) | R (26491) |

| Benchmark | $tilt$ (maxSamp) | Barbarik2 | | |
| --- | --- | --- | --- | --- |
| | | wUniGen (samples) | wSTS (samples) | wQuicksampler (samples) |
| s820a_3_2 | 37 (5e+07) | A (96212) | R (87997) | A (2e+05) |
| s298_15_7 | 44 (6e+07) | A (1e+05) | R (42520) | R (53107) |
| 63.sk_3_64 | 58 (1e+08) | DNS | R (4393) | R (4677) |
| s820a_15_7 | 79 (2e+08) | A (84310) | R (2e+05) | R (16714) |
| s1488_15_7 | 110 (4e+08) | A (86152) | R (17168) | R (7341) |
| s1488_3_2 | 132 (6e+08) | A (89686) | A (89236) | R (7341) |
| s382_15_7 | 138 (6e+08) | A (92159) | R (2e+05) | R (6009) |
| UserServiceImpl.sk_8_32 | 140 (6e+08) | A (1e+05) | R (1e+05) | R (4393) |
| 20.sk_1_51 | 144 (7e+08) | DNS | R (30895) | R (5146) |
| s820a_7_4 | 167 (9e+08) | A (95566) | A (1e+05) | R (6009) |
| s832a_15_7 | 194 (1e+09) | A (96984) | R (9434) | R (13350) |
| s1488_7_4 | 206 (1e+09) | A (1e+05) | R (4677) | R (4627) |
| s344_15_7 | 218 (2e+09) | A (90183) | R (94481) | R (4354) |
| LoginService2.sk_23_36 | 232 (2e+09) | A (1e+05) | R (38044) | R (13350) |
| s420_new1_15_7 | 265 (2e+09) | DNS | R (19224) | A (3e+05) |
| s349_15_7 | 412 (5e+09) | A (99215) | R (28400) | R (14682) |
| s444_15_7 | 501 (8e+09) | A (1e+05) | A (1e+05) | R (26627) |
| s349_7_4 | 603 (1e+10) | A (75555) | R (4284) | R (5150) |
| s444_7_4 | 644 (1e+10) | DNS | R (4393) | R (4354) |
| s420_new1_7_4 | 982 (3e+10) | A (1e+05) | R (4354) | R (18473) |
| s298_7_4 | 986 (3e+10) | A (83681) | R (8638) | R (6009) |
| s420_new1_3_2 | 1226 (5e+10) | DNS | A (1e+05) | R (5150) |
| s382_7_4 | 1283 (5e+10) | A (92307) | R (26491) | R (7341) |

| Benchmark | $tilt$ (maxSamp) | Barbarik2 | | |
| --- | --- | --- | --- | --- |
| | | wUniGen (samples) | wSTS (samples) | wQuicksampler (samples) |
| s420_3_2 | 1552 (8e+10) | A (1e+05) | R (14756) | R (48983) |
| s1238a_7_4 | 1856 (1e+11) | A (95095) | R (5150) | R (7341) |
| s1238a_3_2 | 1965 (1e+11) | A (1e+05) | R (28848) | R (4627) |
| s444_3_2 | 2028 (1e+11) | A (1e+05) | R (2e+05) | R (9500) |
| s1238a_15_7 | 2317 (2e+11) | DNS | R (9020) | R (88233) |
| s420_new_15_7 | 2317 (2e+11) | A (99198) | R (1e+05) | R (4393) |
| 30.sk_5_76 | 2453 (2e+11) | DNS | R (5216) | R (4677) |
| s344_7_4 | 2607 (2e+11) | A (1e+05) | R (14170) | R (16818) |
| s344_3_2 | 3300 (3e+11) | A (1e+05) | R (59952) | R (5150) |
| s420_new_7_4 | 3549 (4e+11) | A (82312) | A (96659) | R (49955) |
| s953a_7_4 | 8984 (3e+12) | DNS | A (2e+05) | R (4627) |
| s953a_15_7 | 10596 (4e+12) | DNS | R (11734) | R (59735) |
| 10.sk_1_46 | 15268 (7e+12) | DNS | R (35179) | R (1e+05) |
| s420_new_3_2 | 17449 (1e+13) | A (1e+05) | R (44937) | R (5150) |
| 19.sk_3_48 | 18253 (1e+13) | DNS | R (59014) | R (4627) |
| s953a_3_2 | 20860 (1e+13) | DNS | R (51161) | R (1e+05) |
| s641_3_2 | 1e+06 (5e+16) | DNS | R (14454) | R (4627) |
| ProjectService3.sk_12_55 | 5e+06 (7e+17) | DNS | R (9020) | R (4393) |
| 71.sk_3_65 | 1e+07 (3e+18) | DNS | R (1e+05) | R (4284) |
| s838_7_4 | 1e+07 (5e+18) | DNS | R (4393) | R (4284) |
| s838_15_7 | 3e+07 (3e+19) | DNS | R (5150) | R (4393) |
| s713_3_2 | 6e+07 (1e+20) | DNS | R (56386) | R (5827) |
| s713_7_4 | 6e+07 (1e+20) | DNS | R (5827) | R (37419) |

| Benchmark | *tilt* (maxSamp) | Barbarik2 | | |
|---|---|---|---|---|
| | | wUniGen (samples) | wSTS (samples) | wQuicksampler (samples) |
| s641_7_4 | 9e+07 (3e+20) | DNS | R (8747) | A (1e+06) |
| s838_3_2 | 2e+08 (1e+21) | DNS | R (9504) | R (4627) |
| 54.sk_12_97 | 4e+11 (6e+27) | DNS | R (14012) | R (4627) |

### D.2 Comparing the runtime performance of Barbarik2 against the baseline approach

In each of the following tables we compare the runtime of Barbarik2 against the runtime of the baseline approach. The runtime of Barbarik2 on `REJECT` instances depends on which iteration the tester terminated on. The runtime of the baseline is extrapolated from the expected number of samples and the average sampling rate of the sampler. To do this we use the $\ell_1$-testing algorithm given in [4]. In the context of this paper, the algorithm assumes black box sample access to a uniform sampler over the models of a Boolean formula $\varphi$, and the sampler under test, and requires $O(\#\varphi^{2/3}(\eta - \varepsilon)^{-8/3} \log(\#\varphi/\delta))$ samples, where $\#\varphi$ is the model count, $(\varepsilon, \eta)$ are the closenes and farness parameters, and $\delta$ is the confidence parameter.

#### D.2.1 Comparision with baseline for wSTS

Table 3: Extended table comparing the baseline tester for wSTS with Barbarik2

| Benchmark | Baseline | Barbarik2(s) | Speedup |
|---|---|---|---|
| s349_7_4 | 16457.21 | 5 | 3428.58 |
| s420_new1_7_4 | 5.4E+6 | 6 | 8.6E+5 |
| s298_7_4 | 705.13 | 8 | 94.02 |
| s444_7_4 | 1.1E+7 | 8 | 1.3E+6 |
| s832a_7_4 | 3725.35 | 10 | 372.53 |
| s1488_7_4 | 184.99 | 12 | 15.16 |
| s344_7_4 | 24751.45 | 15 | 1683.77 |
| s420_3_2 | 2.2E+6 | 17 | 1.3E+5 |
| s1238a_7_4 | 1.4E+6 | 20 | 66538.64 |
| s832a_3_2 | 2149.58 | 22 | 98.60 |
| s832a_15_7 | 15121.66 | 24 | 622.29 |
| s838_15_7 | 2.9E+13 | 27 | 1.1E+12 |
| s349_15_7 | 16457.21 | 28 | 587.76 |
| s838_7_4 | 3.7E+13 | 29 | 1.3E+12 |
| s382_7_4 | 14915.27 | 32 | 469.03 |
| s298_15_7 | 384.62 | 32 | 12.09 |
| s420_new1_15_7 | 4.1E+6 | 33 | 1.3E+5 |
| 27.sk_3_32 | 79531.43 | 34 | 2346.06 |
| s1238a_15_7 | 1.8E+6 | 37 | 49906.05 |
| 111.sk_2_36 | 2.9E+8 | 42 | 6.8E+6 |
| 51.sk_4_38 | 2.0E+6 | 44 | 45904.52 |
| 80.sk_2_48 | 6.0E+7 | 46 | 1.3E+6 |
| s1488_15_7 | 128.69 | 48 | 2.67 |
| s953a_15_7 | 1.1E+9 | 49 | 2.2E+7 |
| s344_3_2 | 15750.92 | 51 | 309.45 |
| s298_3_2 | 229.07 | 52 | 4.42 |
| s838_3_2 | 2.7E+13 | 57 | 4.8E+11 |
| s420_new_3_2 | 2.9E+6 | 65 | 44288.35 |
| 84.sk_4_77 | 3.4E+13 | 68 | 5.0E+11 |
| s641_3_2 | 4.1E+10 | 70 | 5.9E+8 |
| 55.sk_3_46 | 2.0E+7 | 70 | 2.9E+5 |
| s349_3_2 | 30563.39 | 73 | 416.96 |
| 107.sk_3_90 | 1.7E+15 | 86 | 1.9E+13 |
| s1238a_3_2 | 2.2E+6 | 87 | 25824.41 |
| s344_15_7 | 24751.45 | 91 | 271.10 |
| 32.sk_4_38 | 5.8E+5 | 94 | 6228.23 |
| 10.sk_1_46 | 6.5E+7 | 112 | 5.8E+5 |
| 29.sk_3_45 | 2.2E+8 | 150 | 1.5E+6 |
| s420_new_7_4 | 4.1E+6 | 152 | 27272.30 |
| s1488_3_2 | 52.52 | 163 | 0.32 |

Table 3: Extended table comparing the baseline tester for wSTS with Barbarik2

| Benchmark | Baseline | Barbarik2(s) | Speedup |
|---|---|---|---|
| s953a_3_2 | 6.4E+8 | 165 | 3.9E+6 |
| s420_new_15_7 | 4.5E+6 | 186 | 24014.34 |
| 70.sk_3_40 | 2.9E+6 | 201 | 14544.89 |
| s444_15_7 | 13470.05 | 202 | 66.82 |
| s420_new1_3_2 | 2.6E+6 | 211 | 12084.36 |
| s820a_3_2 | 2189.81 | 221 | 9.91 |
| s444_3_2 | 11186.45 | 247 | 45.22 |
| s713_3_2 | 8.8E+10 | 255 | 3.5E+8 |
| 109.sk_4_36 | 6.6E+5 | 269 | 2459.36 |
| s820a_7_4 | 4240.22 | 277 | 15.33 |
| 63.sk_3_64 | 5.8E+11 | 282 | 2.1E+9 |
| s641_7_4 | 8.2E+10 | 311 | 2.6E+8 |
| 53.sk_4_32 | 55060.22 | 313 | 176.08 |
| s382_15_7 | 33182.79 | 343 | 96.86 |
| s820a_15_7 | 4154.77 | 370 | 11.23 |
| ProjectService3.sk_12_55 | 1.3E+10 | 458 | 2.9E+7 |
| s35932_3_2 | 3.6E+2 | TO | - |
| s35932_7_4 | 3.6E+2 | TO | - |
| s35932_15_7 | 3.6E+2 | TO | - |
| s953a_7_4 | 5.7E+8 | 689 | 8.3E+5 |
| UserServiceImpl.sk_8_32 | 479.33 | 720 | 0.67 |
| 30.sk_5_76 | 7.0E+14 | 1116 | 6.2E+11 |
| 77.sk_3_44 | 5.3E+6 | 1687 | 3156.66 |
| tableBasedAddition.sk_240_1024 | 3.8E+14 | 1832 | 2.1E+11 |
| 81.sk_5_51 | 5.0E+9 | 2099 | 2.4E+6 |
| LoginService2.sk_23_36 | 12951.33 | 2368 | 5.47 |
| 20.sk_1_51 | 1.1E+10 | 2568 | 4.1E+6 |
| 19.sk_3_48 | 3.1E+8 | 2760 | 1.1E+5 |
| 17.sk_3_45 | 4.5E+7 | 3016 | 14948.13 |
| 71.sk_3_65 | 4.7E+12 | 4365 | 1.1E+9 |
| 54.sk_12_97 | 2.7E+18 | 4688 | 5.8E+14 |

## D.3 wQuicksampler

Table 4: Extended table comparing the baseline tester for wQuicksampler with Barbarik2

| Benchmark | Baseline | Barbarik2(s) | Speedup |
|---|---|---|---|
| s344_3_2 | 24751.45 | 3 | 8534.98 |
| s344_15_7 | 24751.45 | 4 | 7071.84 |
| s349_7_4 | 28212.36 | 4 | 7624.96 |
| s298_7_4 | 512.82 | 4 | 119.26 |
| s420_new1_3_2 | 5.1E+6 | 4 | 1.2E+6 |
| s420_new_3_2 | 2.2E+6 | 4 | 5.1E+5 |
| s420_new_15_7 | 3.5E+6 | 4 | 7.8E+5 |
| s382_7_4 | 12429.39 | 5 | 2589.46 |
| s444_7_4 | 51980.83 | 5 | 10192.32 |
| s820a_7_4 | 2283.19 | 5 | 430.79 |
| s1488_7_4 | 128.07 | 6 | 20.99 |
| s444_3_2 | 8700.57 | 6 | 1359.46 |
| s838_7_4 | 1.3E+13 | 7 | 1.8E+12 |
| 27.sk_3_32 | 48942.42 | 7 | 6797.56 |
| s1238a_3_2 | 1.6E+6 | 7 | 2.2E+5 |
| s953a_7_4 | 6.6E+8 | 8 | 8.8E+7 |
| s1488_3_2 | 65.65 | 8 | 8.31 |
| s838_3_2 | 1.9E+13 | 8 | 2.4E+12 |
| s1488_15_7 | 60.56 | 9 | 6.80 |
| s349_15_7 | 35265.44 | 9 | 3833.20 |
| s344_7_4 | 22501.32 | 9 | 2393.76 |
| s349_3_2 | 14106.18 | 10 | 1424.87 |
| 55.sk_3_46 | 4.5E+7 | 10 | 4.3E+6 |
| s1238a_7_4 | 1.1E+6 | 11 | 97431.59 |
| s298_3_2 | 534.49 | 11 | 46.89 |
| s832a_7_4 | 4139.28 | 12 | 344.94 |
| 111.sk_2_36 | 5.2E+5 | 12 | 41613.34 |
| s838_15_7 | 2.6E+13 | 12 | 2.1E+12 |
| s420_new1_7_4 | 2.2E+6 | 13 | 1.7E+5 |
| s832a_15_7 | 13861.52 | 14 | 1011.79 |
| UserServiceImpl.sk_8_32 | 326.81 | 14 | 23.68 |
| s382_15_7 | 27149.56 | 15 | 1859.56 |
| 53.sk_4_32 | 91767.04 | 16 | 5595.55 |
| s820a_15_7 | 5665.59 | 17 | 335.24 |
| 84.sk_4_77 | 2.1E+13 | 18 | 1.2E+12 |
| 51.sk_4_38 | 1.8E+6 | 19 | 91363.08 |
| s444_15_7 | 14817.06 | 19 | 763.77 |
| 109.sk_4_36 | 6.6E+5 | 20 | 33425.00 |
| 107.sk_3_90 | 1.6E+15 | 21 | 7.4E+13 |
| 71.sk_3_65 | 1.3E+12 | 27 | 5.0E+10 |
| s641_3_2 | 2.8E+10 | 28 | 1.0E+9 |
| s298_15_7 | 1153.85 | 30 | 38.98 |
| 32.sk_4_38 | 1.2E+6 | 34 | 36689.91 |
| s420_3_2 | 4.5E+6 | 34 | 1.3E+5 |
| s420_new_7_4 | 3.5E+6 | 36 | 96896.05 |
| 80.sk_2_48 | 2.1E+8 | 37 | 5.7E+6 |
| s832a_3_2 | 2149.58 | 45 | 47.66 |
| 19.sk_3_48 | 4.5E+8 | 50 | 9.0E+6 |
| 63.sk_3_64 | 2.1E+11 | 51 | 4.0E+9 |
| 17.sk_3_45 | 8.3E+7 | 55 | 1.5E+6 |

Table 4: Extended table comparing the baseline tester for wQuicksampler with Barbarik2

| Benchmark | Baseline | Barbarik2(s) | Speedup |
|---|---|---|---|
| s713_3_2 | 9.4E+10 | 56 | 1.7E+9 |
| s953a_15_7 | 6.7E+8 | 79 | 8.5E+6 |
| 20.sk_1_51 | 4.0E+9 | 82 | 4.8E+7 |
| 70.sk_3_40 | 4.3E+6 | 101 | 42475.10 |
| s1238a_15_7 | 1.0E+6 | 107 | 9614.31 |
| 10.sk_1_46 | 7.1E+7 | 128 | 5.5E+5 |
| s953a_3_2 | 3.4E+8 | 132 | 2.6E+6 |
| s820a_3_2 | 1167.90 | 137 | 8.54 |
| 30.sk_5_76 | 3.0E+14 | 210 | 1.4E+12 |
| ProjectService3.sk_12_55 | 6.4E+9 | 219 | 2.9E+7 |
| LoginService2.sk_23_36 | 12692.30 | 229 | 55.52 |
| s420_new1_15_7 | 3.2E+6 | 232 | 13726.91 |
| 77.sk_3_44 | 1.2E+7 | 409 | 30125.88 |
| 29.sk_3_45 | 1.3E+8 | 658 | 2.0E+5 |
| 54.sk_12_97 | 4.0E+17 | 690 | 5.8E+14 |
| s641_7_4 | 6.8E+10 | 1117 | 6.1E+7 |
| s35932_15_7 | 1.4E+356 | 1182 | 1.2E+353 |
| tableBasedAddition.sk_240_1024 | 3.0E+13 | 1430 | 2.1E+10 |
| s35932_7_4 | 1.2E+356 | 2227 | 5.5E+352 |
| s35932_3_2 | 1.1E+356 | 2346 | 4.5E+352 |
| 81.sk_5_51 | 2.0E+9 | 2461 | 8.3E+5 |

## D.4  wUniGen

Table 5: Extended table comparing the baseline tester for wUniGen with Barbarik2

| Benchmark | Baseline | Barbarik2(s) | Speedup |
|---|---|---|---|
| s1488_3_2 | 229.78 | 6648 | 0.03 |
| s298_7_4 | 7564.11 | 10758 | 0.70 |
| s1488_15_7 | 643.45 | 11493 | 0.06 |
| s298_15_7 | 2948.72 | 12325 | 0.24 |
| s349_7_4 | 1.8E+06 | 12858 | 136.40 |
| s820a_15_7 | 48724.11 | 14070 | 3.46 |
| s344_15_7 | 3.8E+05 | 14074 | 27.18 |
| s1488_7_4 | 853.78 | 15049 | 0.06 |
| s820a_7_4 | 42728.33 | 16124 | 2.65 |
| s349_15_7 | 3.9E+05 | 17690 | 21.80 |
| s382_7_4 | 9.7E+05 | 21785 | 44.45 |
| s349_3_2 | 3.0E+05 | 22395 | 13.54 |
| s832a_15_7 | 5.6E+05 | 23036 | 24.45 |
| s420_new_7_4 | 4.0E+09 | 24092 | 1.7E+5 |
| s344_7_4 | 1.7E+06 | 26423 | 64.55 |
| 51.sk_4_38 | 2.7E+09 | 26612 | 1.0E+5 |
| s820a_3_2 | 2.3E+05 | 27408 | 8.47 |
| s298_3_2 | 2061.62 | 30262 | 0.07 |
| s344_3_2 | 5.0E+05 | 32378 | 15.29 |
| s1238a_7_4 | 1.5E+09 | 33689 | 45408.69 |
| s832a_7_4 | 76990.55 | 34315 | 2.24 |
| s382_15_7 | 1.0E+07 | 39024 | 263.98 |
| s1238a_3_2 | 7.1E+08 | 40406 | 17575.38 |
| s420_new_15_7 | 4.9E+09 | 40725 | 1.2E+5 |
| 27.sk_3_32 | 7.4E+06 | 41997 | 176.26 |
| s832a_3_2 | 74844.43 | 42696 | 1.75 |
| UserServiceImpl.sk_8_32 | 21547.88 | 45090 | 0.48 |
| 32.sk_4_38 | 4.9E+08 | 45126 | 10872.88 |
| s420_new1_7_4 | 2.8E+08 | 48911 | 5639.38 |
| s444_3_2 | 1.9E+06 | 55017 | 34.61 |
| LoginService2.sk_23_36 | 1.3E+06 | 56229 | 22.38 |
| s420_3_2 | 2.3E+09 | 68048 | 33247.50 |
| 53.sk_4_32 | 2.2E+07 | 70590 | 312.87 |
| s420_new_3_2 | 1.2E+10 | 75284 | 1.6E+5 |

## D.5 Number of samples required for baseline approach

Table 6: Number of samples required for baseline tester

| Benchmark | Number of samples |
|---|---|
| s344_3_2 | 2E+6 |
| s344_15_7 | 2E+6 |
| s349_7_4 | 2E+6 |
| s298_7_4 | 6E+4 |
| s420_new1_3_2 | 3E+8 |
| s420_new_3_2 | 3E+8 |
| s420_new_15_7 | 3E+8 |
| s382_7_4 | 1E+6 |
| s444_7_4 | 4E+6 |
| s820a_7_4 | 3E+5 |
| s1488_7_4 | 1E+4 |
| s444_3_2 | 1E+6 |
| s838_7_4 | 2E+15 |
| 27.sk_3_32 | 6E+6 |
| s1238a_3_2 | 1E+8 |
| s953a_7_4 | 4E+10 |
| s1488_3_2 | 7E+3 |
| s838_3_2 | 2E+15 |
| s1488_15_7 | 8E+3 |
| s349_15_7 | 2E+6 |
| s344_7_4 | 2E+6 |
| s349_3_2 | 2E+6 |
| 55.sk_3_46 | 2E+9 |
| s1238a_7_4 | 1E+8 |
| s298_3_2 | 4E+4 |
| s832a_7_4 | 4E+5 |
| 111.sk_2_36 | 3E+7 |
| s838_15_7 | 2E+15 |
| s420_new1_7_4 | 3E+8 |
| s832a_15_7 | 1E+6 |
| UserServiceImpl.sk_8_32 | 2E+4 |
| s382_15_7 | 3E+6 |
| 53.sk_4_32 | 6E+6 |
| s820a_15_7 | 4E+5 |
| 84.sk_4_77 | 1E+15 |
| 51.sk_4_38 | 8E+7 |
| s444_15_7 | 1E+6 |
| 109.sk_4_36 | 4E+7 |
| 107.sk_3_90 | 7E+16 |
| 71.sk_3_65 | 5E+13 |
| s641_3_2 | 3E+12 |
| s298_15_7 | 6E+4 |
| 32.sk_4_38 | 7E+7 |
| s420_3_2 | 3E+8 |
| s420_new_7_4 | 3E+8 |
| 80.sk_2_48 | 6E+9 |
| s832a_3_2 | 2E+5 |
| 19.sk_3_48 | 1E+10 |
| 63.sk_3_64 | 5E+12 |

Table 6: Number of samples required for baseline tester

| Benchmark | Number of samples |
|---|---|
| 17.sk_3_45 | 2E+9 |
| s713_3_2 | 6E+12 |
| s953a_15_7 | 5E+10 |
| 20.sk_1_51 | 7E+10 |
| 70.sk_3_40 | 2E+8 |
| s1238a_15_7 | 1E+8 |
| 10.sk_1_46 | 6E+9 |
| s953a_3_2 | 4E+10 |
| s820a_3_2 | 1E+5 |
| 30.sk_5_76 | 2E+15 |
| ProjectService3.sk_12_55 | 2E+11 |
| LoginService2.sk_23_36 | 1E+5 |
| s420_new1_15_7 | 3E+8 |
| 77.sk_3_44 | 3E+8 |
| 29.sk_3_45 | 3E+9 |
| 54.sk_12_97 | 4E+18 |
| s641_7_4 | 5E+12 |
| s35932_15_7 | 1E+357 |
| tableBasedAddition.sk_240_1024 | 1E+15 |
| s35932_7_4 | 1E+357 |
| s35932_3_2 | 1E+357 |
| 81.sk_5_51 | 4E+10 |

## Footnotes

[3] for any $\varepsilon < \frac{1}{3}$ and $\eta > 6\varepsilon$

[4] for any $\varepsilon < \frac{1}{3}$ and $\eta > 6\varepsilon$

[5]$H \geq L$ if $hi \geq lo$ that is $(6\varepsilon + \eta)/4 \geq (2\varepsilon)/(1-\varepsilon)$