[Reviews · NeurIPS 2020]

Review 1

Summary and Contributions: This paper presents a new algorithm, called Fulcrum, for testing general-purpose tools that perform constrained sampling of discrete distributions. Fulcrum applies to samplers that accept any Boolean formula and any target distribution on the satisfying assignments of the formula, rather than to algorithms designed for sampling particular distributions. For such a sampler G, and a particular query formula and weight function, Fulcrum attempts to determine either that: 1) The sampler is multiplicatively close (i.e., (1 – eps)p(x) < G(x) < (1 + eps)p(x)) to an ideal sampler for the target distribution in question; or 2) The sampler’s distribution is far in L1-norm from an ideal sampler for the target distribution in question. Fulcrum’s sample efficiency is a function of the tilt of the target distribution, i.e., the ratio between the probabilities of the highest- and lowest-weight satisfying assignments. The key to Fulcrum’s efficiency is that because the samplers it is testing are algorithms that can be applied to any query, Fulcrum can generate derived queries that allow it to test the sampler’s behavior on conditioned versions of the original query. In particular, suppose that an ideal sampler and the sampler G under test produce different samples. Fulcrum can construct a new target distribution over only these two satisfying assignments, and test whether G is unduly biased toward the one that it produced. This is essentially how the algorithm works: a handful of samples are generated both from a slow ideal sampler and from G, and whenever the samples are not equal, a bias test is performed. If any of the bias tests fail, G is rejected. If they all pass, G is accepted. In order for these derived tests to be informative about the sampler’s behavior on the original query, Fulcrum must assume a property the authors call “non-adversariality.” If a sampler is “non-adversarial,” it means that when Fulcrum constructs a derived query that restricts the solution space of some original query, the sampler behaves like a rejection sampler, simulating according to its original sampling distribution but restricted to smaller solution set. In other words, a “non-adversarial” sampler is consistent in the approximations it makes: if its behavior on some query (phi, wt) is wrong, then its behavior on a derived query that further restricts the space of solutions will be wrong in the same way. The paper tests Fulcrum on three samplers, all of which were originally developed as tools for sampling (approximately) uniformly from the satisfying assignments of Boolean formulas. These tools are adapted by the authors to serve as samplers for a certain class of non-uniform distributions. In particular, these are distributions arising from independent products of Bernoulli variables with potentially unequal weights, conditioned on a Boolean predicate (Appendix D). Such queries can be transformed approximately into queries for uniform samplers, using the “inverse transform” trick. On a suite of benchmarks, Fulcrum generally rejects the two samplers that lack theoretical guarantees, and accepts the sampler (wUniGen) that does come with guarantees.

Strengths: Fulcrum is a neat algorithm. I have not checked all the proofs in detail, but I did not find any bugs in the claims. I like the observation that a sampler can be queried not just with the distribution under test but also conditioned versions of that distribution, to quickly gather information about whether the sampler is behaving properly on a projection of the original problem. Fulcrum also seems straight-forward to implement. It also seems useful that when Fulcrum rejects a sampler, it provides a witness of two assignments that the sampler generates in incorrect proportions. I don’t know whether property-based testing typically requires non-uniform sampling, but it seems plausible to me that practitioners in that field may find it valuable to see the Barbarik algorithm of [10] extended to the non-uniform case. The machine learning community is of course very interested in algorithms for sampling complex discrete distributions, and in finding efficient ways of testing when these (usually approximate) algorithms actually work. To the extent that Fulcrum paves the way for more research on this question, it could be of great value to the NeurIPS community.

Weaknesses: It is hard for me to evaluate the significance of Fulcrum and its relevance to the NeurIPS community. The tools to which Fulcrum is applied in this paper are from the software engineering / property-based testing literature; they are intended for generating uniform samples for software testing, but modified by the authors for generating from a limited set of non-uniform distributions. It is certainly possible that this weighted sampling task is of interest to the property-based testing community, but I don’t know enough to say. So my perspective is limited: I can comment only on Fulcrum’s apparent applicability to problems of interest to a broader NeurIPS audience. To that end: do the approximate sampling algorithms that machine learning practitioners and statisticians regularly use, and which this paper mentions as motivation (e.g., MCMC with a finite chain length, simulated annealing, variational inference), plausibly satisfy the ‘non-adversarial’ condition (or something close to it)? My worry is that a particular MCMC algorithm’s behavior on one target distribution may not be particularly informative of its behavior of another. (Indeed, many MCMC algorithms are designed only to work on particular target distributions, with custom Markov chains constructed to exploit their structure; in these cases, it is unclear if or how Fulcrum could be applied.) Fulcrum’s empirical evaluation is very nice, but as far as I can tell does not apply Fulcrum to samplers or target distributions that have typically been of interest to the ML/Stats community. Because of this, the paper doesn’t seem to quite deliver on the stated motivation in Section 1.

Correctness: I believe so.

Clarity: These are all relatively minor things, but I found the development unclear at parts. For example, the notion of a "Chain formula" is used without definition (I later realized there is a definition in the appendix), and the notation in Definition 8 was confusing: the down arrow is used elsewhere to restrict to a particular set of variables, but in this definition, it is used to mean "conditioning on an event" (where T is the event). This is not explained. Kernels are defined as deterministic functions, but you write that Kernels can be designed with "enough randomization" to admit many samplers as non-adversarial -- what exactly does this mean? What does it mean for phi hat to have "similar structure" to phi? I think it is possible to understand the paper despite these unclear parts, but it did make it more difficult (and I am not 100% sure I have correctly understood). Several terms were not used the way I'd expect them to be used. For example, I wouldn't expect the "support" of a distribution to refer to the set of variables over which it is defined, but rather the set of assignments to those variables with non-zero probability. Furthermore, suggestive names like "non-adversarial" are used but are not justified. (I realize this definition comes from another paper. But is it a good name for this property? It seems many samplers will not satisfy it, despite not being constructed adversarially.)

Relation to Prior Work: I am not especially familiar with related work in this field, but the authors do explain how Fulcrum differs from what appear to be the most similar methods. And Fulcrum seems quite different from the more traditional "goodness-of-fit" tests in statistics.

Reproducibility: Yes

Additional Feedback: Post response: =========== Thanks for taking the time to put together this response. I am revising my score to recommend accepting the paper, as I have been convinced by R2 that it is an exciting approach that could inspire follow-on work addressing some of its current weaknesses. I would strongly encourage the authors, however, to include a more careful discussion of the non-adversariality condition in a revision. I disagree with the statement in the author response that it amounts to only "support for conditioning." It is in fact much stronger: it requires that sampling algorithms are *consistent* in the approximations they make, i.e., the relative probability that a sampler produces x vs. y does not change when the input distribution is altered to condition on an event. I would not expect algorithms like MCMC and simulated annealing, which rely on local search, to have this property: for example, simulated annealing may work well on a broad and unconstrained target, while failing to explore a narrow target that has been conditioned on a rare event. One possible name for "non-adversariality" that arose during post-review discussion was "subquery consistency": this emphasizes that a sampler's behavior must be consistent when invoked with subqueries (conditioned versions) of the original query. Even if the authors do not change the name, I do think that more discussion of the restrictions imposed by the assumption would be useful for readers.


Review 2

Summary and Contributions: The paper introduces an efficient testing method for samplers targeting discrete distributions. It can be used to compare any sampler capable of sampling propositional assignments for a given weighting function and an arbitrary constraint predicate to a reference sampler known to be correct. The approach is to draw one assignment from the tested sampler and one from the reference sampler, extend the constraint in a way that only admits those two assignments, and repeatedly sample from the tested sampler to check if the ratio of frequencies of the two assignments matches the ratio of their weightings. The authors give probabilistic guarantees for the power of their testing method, both for accepting similar samplers and rejecting different ones, with the number of samples needed constant in the size of the sample space. The guarantee is proven under the assumption of the tested sampler being non-adversarial, which roughly means that adding further constraints doesn't change the relative probabilities of sampling satisfying assignments. While this is generally difficult to ensure, the paper also presents a heuristic method to obfuscate the constraint and make it difficult for the sampler to behave adversarially, which is shown to work well empirically. EDIT: I stand by my initial assessment that this is a great paper introducing some very original and powerful ideas. One thing I would like to point out is that, as discussed by R1, the non-adversarial assumption is quite strong and it is rather misleading to label it as "non-adversarial", since it is likely to be violated by many existing approximate samplers without anyone intending for them to do so. I understand that this term comes from existing literature, but given how big of a topic adversarial examples and adversarial networks are at NeurIPS, using this term here is very likely to confuse a lot of people. I would strongly recommend that the authors find a different term to describe this property, my suggestion would be "subquery-consistent".

Strengths: Efficient testing of random samplers is a very important practical problem which doesn't currently have any good solutions, so the impact of this work is potentially very high. Although this is not my area of expertise, I believe the presented approach is substantially novel and could inspire a lot of follow-up work. The algorithm is presented very clearly in a rigorously defined setting, and the correctness proves provided in the appendix appear sound, although I have not checked them carefully. Empirical evaluation makes sense, the hypotheses being tested are clearly stated and the results are discussed as they relate to the hypotheses. I do not have the expertise to comment on the choice of benchmarks. Supplementary material includes code for running the experiments, although I have not tried using it.

Weaknesses: One thing I was hoping to find in the empirical evaluation is an ablation that doesn't use the constraint obfuscation heuristic to see if the naive approach would cause the samplers to behave adversarially. Also the last sentence of the conclusion mentions that the method can't address all possible discrete distributions, but it's not clear exactly what types of distributions would not work.

Correctness: The claims are correct and the proofs are sound. The empirical methodology makes sense.

Clarity: The paper is clear, with definitions and the full algorithm stated in main text, while proofs and derivations are delegated to the appendix. The main text discusses the key issues and avoids the relatively standard details.

Relation to Prior Work: The paper makes it clear how this work overcomes the shortcomings of existing approaches.

Reproducibility: Yes

Additional Feedback:


Review 3

Summary and Contributions: The paper aim to compare sampling schemes and propose an algorithm named Fulcrum to perform hypothesis tests between samplers. The paper stats a bound characterizing the distance, in terms of a measure based on empirical distribution function, upon which the rejection decision is made.

Strengths: It is certainly an interesting problem setting to compare and criticize different sampling schemes. However, it is not super clear to me which aspect of the sampling scheme the paper aims to address/criticize.

Weaknesses: The explanation of the problem setting is not super clear to me. To illustrate, please refer to the following questions: 1.Do we have a ground through of distribution we would like to sample? Or in another words, if you already have a ideal sampler, are you testing a different sampling scheme is approximately good? 2. The hypothesis testing part is not clear. It states that the the statistics based on ideal sampler is bounded by some parameter related to \delta. (In line 244) how does \delta relate to test size (Type I error)? 3. what does the kernel mean here, how does it compare/realte to RKHS kernel. 4. if we have ground truth distribution, is this consider as ideal sampler? if so, what is the advantage of comparing sampler different from a goodness-of-fit test setting? If the ground truth is in sample form, how does the proposed setting compare with the two-sample test setting? 5. Is there any results related to test power? (1-Type II error)

Correctness: It claims on the results for hypothesis testing, where the distribution under the null and the alternative hypothesis is yet clearly stated. The rejection region is not convincing.

Clarity: There are certain aspects of clarity to be improved for the reader to better understand the results.

Relation to Prior Work: The related work are well stated.

Reproducibility: Yes

Additional Feedback: ======post review==== Thanks for then clarifying the setting and help me better aware of the topic. I now better understand the testing criterion and procedure. I agree with other reviewers on the merit of this work.


Review 4

Summary and Contributions: This work proposes an algorithm to testing samplers i.e. verifying whether a sampler generates samples according to a correct target distribution. --UPDATE -- I have read the author's response and other reviews. As a "proof of concept", I do this work has a potential to motivate future work as it is an example of a cross-pollination of different fields. As such I increase my score to 7.

Strengths: Providing a solution for a relatively novel challenge as well as theoretical analysis of its complexity are the strength points of this work.

Weaknesses: This method (as it is) is only applicable to discrete distributions, the experiments are minimal and the proven bounds are not tight: According to the presented theorem, the upper bound on the required samples for the tester, i.e. the complexity of the proposed tester depends on a quantity, namely "tilt" (see definition 4), which can be arbitrary large (and in many cases, unknown). In the absence of a tight and practical bound, it seems that the problem of evaluation of the performance of samplers are transferred into the problem of evolution of the sampler tester which seems at least equally challenging.

Correctness: I did not follow all details but as far as I can say, the method seems correct.

Clarity: The paper is well written however, if the provided definitions where accompanied by a simple running example, it could make their understanding much easier.

Relation to Prior Work: Yes, the prior work and their relation to this work are presented reasonably well.

Reproducibility: Yes

Additional Feedback:

[Author Response · NeurIPS 2020]

We are deeply appreciative of reviewers for their feedback amidst these trying circumstances. We are glad that reviewers appreciated the novelty of the approach for a fundamental problem, the rigor of our analysis, simplicity of the approach from an implementation perspective.

[R3] There seems to be a serious misunderstanding, perhaps due to certain terminology used in our paper. In this paper, we are indeed interested in comparing different sampling schemes with respect to the number of samples required. Since we are interested in discrete distributions, most standard sampling schemes, like the two-sample scheme, can be used. But such schemes suffer from an impractical requirement on the number of samples. The tests in hypothesis testing literature operate in a *black-box setting*, wherein we have strong lower bounds [4,36,37] over the required number of samples before a statistically sound conclusion can be drawn. Our setting is not black-box, as pointed out in the Abstract and Introduction. To borrow from folklore in property testing, consider two distributions $p$ and $q$ such that $p$ is the uniform distribution over an $N$ element set $S$ and $q$ is either equal to $p$ or is the uniform distribution over a subset $T$ of $S$ where $|T| = \frac{N}{2}$, that is for every element in $T$ the probability of that element in $q$ is $\frac{2}{N}$. If the set $T$ is unknown, the number of samples one would need to check if $q$ is equal to or far from the uniform distribution $p$ over the set $S$ is at least $\sqrt{N}$. Since $N$ is exponential in the number of bits, the sample complexity has an exponential lower bound. This lower bound is exhibited in [4]. In our setting, since N is exponential in the input size, the sample complexity is also exponential. As mentioned in the Introduction, the usage of conditional sampling allows us to circumvent the hardness, and this work shows how we can handle arbitrary discrete distributions.

We refer the reader to citations [8] and [9] for detailed discussions about property testing literature and the limitations of the standard hypothesis testing setting. While we are aware of the literature on various sampling schemes from hypothesis testing, our paper does not exactly fit in that line of work and instead fits the property testing line of work, as pointed out in the Appendix. Therefore, we did not attempt to cast our results in the notions of null and alternate hypothesis, even though such a characterization is possible but non-standard in property testing literature. Furthermore, we are not sure about the need for such a formulation when Definition 6 and Theorem 1 are self-contained and precise.

[R3] We have access to an Ideal sampler in addition to the weight function (lines 80-85). An explicit description of ground truth distribution in terms of samples is intractable given the large *model count* (see Appendix E2).

[R1, R3] **About the Kernel and the role of randomization.** Kernel, as defined in Definition 7, allows the usage of randomization internally. As discussed on line 229, we use randomization merely to choose the literal on line 6 of Algorithm 2; we will add further discussion to this effect in the final version (We have provided source code with a detailed description). The Kernel in our paper has nothing to do with RKHS. Perhaps we should use a different name to avoid confusion.

[R4] As is the norm, we focused on binary variables, but the techniques work for any arbitrary discrete domain.

[R1, R4]: As a first step to demonstrate the working of the prototype implementation, we focused on log-linear models for which the most efficient technique is to employ the inverse transform. This also allows us to easily obtain a sampler with formal guarantees and two other samplers without guarantees. It is worth noting that STS and variants of UniGen have indeed appeared in AI/ML conferences, so we have taken the first step in addressing the distributions of interest to the AI/ML community. We do agree with [R2] (and also strongly believe) that our algorithmic framework will inspire follow-up work focused on further improvements and a deeper understanding of various complex distributions. Again, our algorithmic analysis is not restricted to log-linear models.

[R1] **Do sampling algorithms satisfy the 'non-adversarial' condition ?** Since conditioning is a fundamental operation in probabilistic reasoning, one would expect that the underlying sampling algorithms support conditioning. Our non-adversariality assumption simply translates to support for conditioning. We will emphasize this further. At the same time, we do agree with the reviewer that one can claim that their sampler supports only one particular distribution, but in such a case, it is hard to see if one can do any better given the strong lower bounds [4,36,37].

[R1] **What does it mean for $\hat{\varphi}$ to have "similar structure" to $\varphi$?** $\hat{\varphi}$ is constructed by conjuncting $\varphi$ with a small formula which allows conditional sampling on the models of $\varphi$; thus it can be said that $\hat{\varphi}$ retains the structure of $\varphi$.

[R2] **What is the performance like without the constraint obfuscation heuristic, does the naive approach work?** Our preliminary studies showcased the need for constraint obfuscation methodology. We will seek to conduct a longer empirical study and add these results in the final version.

[R4] **The absence of a tight and practical bound.** It is worth emphasizing that Theorem 1 only implies the worst-case behavior of Fulcrum, and lack of tight upper bound does not imply that the performance of Fulcrum would be worse, a claim supported by experimental analysis as well. As a similar example, while there are no bounds better than exponential time for any algorithm to solve SAT, the SAT solvers routinely perform orders of magnitude better than a naive exponential time algorithm. In this view, the design of an efficient algorithm should be viewed as a primary contribution even though its worst-case behavior is hard to analyze.

[Meta-Review · NeurIPS 2020]

This paper introduces a method for testing samplers with discrete distributions as targets. The authors demonstrate its usefulness in theory and experiments. The reviewers seem to be in agreement that the main strength of the paper is its novelty in addressing an important problem (testing of samplers). While the presented method does not (as laid out in this paper) apply to typical samplers used for approximate inference used by the sampling community at NeurIPS, the reviewers are excited by the potential for the novelty of this approach to inspire further work at the intersection of these areas. The authors should make sure to address the reviewers' concerns about the naming and discussion of the "non-adversariality" condition in their camera-ready version. See the updated reviews from Reviewers 1 and 2. In particular, the authors should be sure to rename this condition and discuss its restrictions more fully in their final paper.